# The neurobiological basis of affect is consistent with psychological construction theory and shares a common neural basis across emotional categories

Doğa Gündem [1], Jure Potočnik[1], François-Laurent De Winter[1,2], Amal El Kaddouri[1], Daphne Stam [1], Ronald Peeters[3], Louise Emsell [1,3,4], Stefan Sunaert[3,4], Lukas Van Oudenhove [5,6], Mathieu Vandenbulcke[1,2], Lisa Feldman Barrett [7,8,9] & Jan Van den Stock [1,2✉]

Affective experience colours everyday perception and cognition, yet its fundamental and neurobiological basis is poorly understood. The current debate essentially centers around the communalities and specificities across individuals, events, and emotional categories like anger, sadness, and happiness. Using fMRI during the experience of these emotions, we critically compare the two dominant conflicting theories on human affect. Basic emotion theory posits emotions as discrete universal entities generated by dedicated emotion category-specific neural circuits, while psychological construction theory claims emotional events as unique, idiosyncratic, and constructed by psychological primitives like core affect and conceptualization, which underlie each emotional event and operate in a predictive framework. Based on the findings of 8 a priori-defined model-specific prediction tests on the neural response amplitudes and patterns, we conclude that the neurobiological basis of affect is primarily characterized by idiosyncratic mechanisms and a common neural basis shared across emotion categories, consistent with psychological construction theory. The findings provide further insight into the organizational principles of the neural basis of affect and brain function in general. Future studies in clinical populations with affective symptoms may reveal the corresponding underlying neural changes from a psychological construction perspective.

[1] Neuropsychiatry, Department of Neurosciences, Leuven Brain Institute, KU Leuven, Leuven, Belgium. [2] Geriatric Psychiatry, University Psychiatric Center KU Leuven, Leuven, Belgium. [3] Department of Radiology, University Hospitals Leuven, Leuven, Belgium. [4] Department of Imaging and Pathology, KU Leuven, Leuven, Belgium. [5] Laboratory for Brain-Gut Axis Studies (LaBGAS), Translational Research in Gastrointestinal Disorders (TARGID), Department of Chronic Diseases and Metabolism, Leuven Brain Institute, KU Leuven, Leuven, Belgium. [6] Cognitive and Affective Neuroscience Lab, Department of Psychological and Brain Sciences, Dartmouth College, Hanover, NH, USA. [7] Department of Psychology, Northeastern University, Boston, MA, USA. [8] Department of Psychiatry, Massachusetts General Hospital, Harvard Medical School, Boston, MA, USA. [9] Athinoula A. Martinos Center for Biomedical Imaging, Massachusetts General Hospital, Charlestown, MA, USA. ✉email: jan.vandenstock@kuleuven.be

The fundamental structure and biological basis of human emotions has incited human interest for millennia[1,2]. Stringent and less stringent variants of basic emotion theory (BET) have dominated the field in the last decades[2–4]. According to BET, there are a limited number of basic emotion categories such as anger, sadness, happiness, fear, disgust, and surprise that differ in expression, appraisal, physiology, and behavioral response[3]. BET posits that basic emotions have a universal nature and are thus consistent across individuals and cultures. A recent large-scale study of YouTube videos revealed that associations of 16 facial expression dynamics with specific contexts showed a 70% world-wide consistency[5]. Emotion categories are considered natural kinds, neurobiologically hard-wired, and inherited with a dedicated neural circuit for each basic emotion category[6–8]. BET claims that emotions arise from integrated neural circuitry including the brain stem, amygdala, insula, anterior cingulate, and orbitofrontal cortices[9]. In particular, the amygdala is typically associated with fear[10,11], the anterior insula with disgust[12,13], the orbitofrontal cortex (OFC) with anger[14,15], the anterior cingulate cortex (ACC) with happiness[15] and medial prefrontal cortex (MPFC) with sadness[15]. The hypotheses that each event of a particular emotion category is processed via its specific neurobiological architecture and that this architecture is similar across subjects are at the core of BET.

Psychological construction theories (PCT) of emotion propose an anti-essentialist approach and claim that while each emotional event is unique, a common set of fundamental psychological operations underlies the processing of every emotional event. These operations include core affect (reflecting valence and arousal) and conceptualization (generating meaning by integrating external with internal signals via associations with past experiences)[16–20]. Each of these common underlying functions is neurobiologically supported by large-scale networks, such as the salience and default mode network[16–19,21] and intrinsic allostatic-interoceptive brain systems[22]. The psychological primitives construct emotions, operating on a predictive basis[22]. As personal history and cultural factors are important determinants of prediction generation, the emphasis in PCT is on idiosyncrasy and cultural specificity rather than on universality[23]. Hence, similarities across and specificities between emotion categories constitute core distinctions between BET and PCT.

Meta-analytic studies on the neurobiological basis of emotions have yielded mixed results, with support for BET reflected by the finding of only partially overlapping arrays of structures between distinct emotion categories[15,24], support for PCT reflected in distributed functional clusters that are consistently activated across emotions[19,25] and a flexible set of limbic and paralimbic brain regions supporting valance-general responsivity as large-scale brain activity[20] or inconclusive results[14,26].

Importantly, meta-analyses on emotional processing typically make abstraction of the phenomenological quality of emotional processes like perception and experience, despite the evident differences between them. One of the main qualifications of emotion processing relates to perception vs experience ('affect') of emotion. The vast majority of emotion studies have investigated emotion perception, typically conveyed by facial expressions. An important advantage of perception studies is that emotional stimulation can be highly standardized. However, an underlying and often implicit assumption of theorists that propose emotion models based on emotion perception studies, is one of uni-potentiality, i.e. the notion that sensory stimulation with for instance an angry face can only result in an anger response in the observer. However, it is clear that both external and internal factors influence how emotional expressions are interpreted and thus the control over the emotional response in the observer is limited[27–29]. The predominantly behavioural orientation of psychology and affective neuroscience in the past decades ensued in an under-exploration of the neural basis of affect, by definition a highly subjective process[30]. In line with this, the evidence favouring PCT is largely based on emotion perception studies[31,32].

The aim of the present study is to focus on the neural basis of affect and critically test specific predictions derived from BET and PCT. We induce affect via the recollection of autobiographical events[33,34] and combine conventional group-level analyses with tailored methodological approaches to addresses model-specific hypotheses regarding specificity and consistency of effects relating to emotional events, emotional categories, and subjects. Furthermore, we investigate this at the level of the amplitude and the pattern of the neural response[35–37]. In particular, we investigate 8 specific predictions centred around 3 topics.

First, we test between emotion category specificity. BET predicts high specificity in neural architecture between distinct emotion categories[7,38,39]. Stringent and simplistic variants of BET propose a one-to-one mapping between emotion categories (e.g. anger) and activation in well-defined neural structures (e.g. orbitofrontal cortex, OFC). More contemporary BET variants acknowledge the modulation of distant regions by core structures as well as the importance of the neural pattern in addition to the intensity of the activation[40]. In this perspective, a recent study reported that 12 out of 14 emotion categories including basic and non-basic emotions were distinguishable based on neural pattern, claiming that different emotions can be characterized by distinct neural signatures within a shared neural circuitry[41]. PCT anticipates lower specificity between emotion categories based on the large-scale brain circuits that underlie the psychological ingredients that are shared between all mental states and the postulated neural complexity through degeneracy. The latter refers to the characteristic that different sets of neurons can underlie the processing of events of a single category, reflecting a many-to-one mapping of structure and function[42]. Here, we investigate emotion category-specific activation via contrasting each emotion category with each of the remaining categories. For between category specificity, we therefore consider the following specific hypothesis: 'H1: Each emotion category activates dedicated structures compared to any other category (BET)'.

Secondly, we investigate within emotion category consistency. BET comprises explicit phylogenetic hypotheses about emotion category-specific neural circuitries as well as on the universal characteristics of basic emotions, including their neural basis. The essentialist assumption that each event of a particular basic emotion is processed via the category-specific neural circuit, implies the notion that neural signals during different events of a single emotion category will display at least a minimal amount of overlap in activation topography and/or a minimal similarity in activation pattern across events and across subjects. Consistency in neural activation and/or activation pattern across events as well as across subjects for a single emotion category is one of the key predictions of BET. PCT predictions on consistency across events are less unambiguous. On the one hand, a high overlap and/or similarity can be anticipated based on the common networks shared across different emotion events within a single emotion category. On the other hand, the relative contribution of a network may vary substantially across events and across subjects, based on the proposed high degree of specificity of each event and importance of ontogeny and idiosyncrasy in PCT. Therefore, the anticipated consistency of neural activation topography and activation patterns across subjects is lower in PCT compared to BET. Four specific hypotheses are considered in this context: 'H2: Different events of a single emotion category activate similar structures (BET)', 'H3: The similarity between neural patterns across events within an emotion category (vs neutral) is

significantly higher than between emotion category (vs neutral) (BET)', 'H4: There is a significant overlap in regional activation across subjects within an emotion category (BET)', and 'H5: The similarity between neural patterns across subjects within an emotion category is significantly higher than between emotion category (BET)'.

Finally, we investigate across emotion category consistency. A core distinction between BET and PCT relates to similarities in neural signals between emotion categories. While categorical specificity is a hallmark of BET, the reverse applies to PCT. Indeed, PCT hypothesizes a common set of psychological functions and large-scale neural networks that underlie the processing of each event, either emotional or neutral. These systems operate regardless of emotion category. Therefore, we investigated between category topographic activation overlap and pattern similarities at the group level and interindividual level, resulting in the following specific hypotheses: 'H6: The overlap in activation between emotion categories is high (PCT)', 'H7: There is a significant association between emotion categories across subjects (PCT)', and 'H8: The similarity between neural patterns across events within an emotion category (vs baseline) is significantly higher than between emotion categories (vs baseline) (BET)'.

To test these predictions, we made use of a paradigm that differs from conventional emotion processing paradigms in 2 aspects. First, the protocol aims to maximize the match across subjects of the emotional response. This differs from mainstream emotion processing studies in which the same emotional stimuli are presented to all subjects and it is assumed that these stimuli trigger similar emotional responses across all subjects. Secondly, the protocol is optimized to obtain reliable estimates of each event at subject level without increasing the probability of psychological and neural fatigue/adaptation effects. Furthermore, the paradigm includes phenomenological variation within category between events, and we performed this in a sample of 37 healthy subjects. Finally, by instructing the subjects to close their eyes, we were able to minimize the influence of external visual input and control for looking preferences and fixations across subjects. Participants also indicated the experienced emotional intensity.

We used an independent dataset to define a general affect network (GAN)[20] and compared this to the areas that were modulated by emotion in our study. We also focused on primary BET regions associated with anger, sadness, and happiness, i.e. OFC, MPFC, and ACC respectively. We tested 8 a priori-defined predictions from emotion theories centred around 3 topics across the whole brain, within the GAN, and emotion associated BET regions. The results revealed low between emotion specificity and within emotion consistency, conflicting with BET, and high across emotion consistency, compatible with PCT. We conclude that the neurobiological basis of affect is characterized by biological primitives underlying multifarious emotional events supported by a large-scale network, in line with PCT.

## Results

**Behavioural results.** The intensity ratings averaged over events of each emotion category ranged between 0.16 and 1 (on the scale of 0–1 as from 'very weak' to 'very intense'). Shapiro-Wilk tests revealed that intensity ratings of anger were not normally distributed (anger: $p = 0.034$; sadness: $p = 0.056$; happiness: $p = 0.265$). Kruskal-Wallis tests revealed that intensity ratings did not differ between emotion categories ($\chi^2(2) = 3.291$, $p = 0.193$). Furthermore, Shapiro-Wilk tests revealed that not all the intensity ratings across subjects within an event of an emotion were normally distributed (all $p$'s > .0005). Wilcoxon signed rank tests revealed that the intensity ratings of the emotion experience between events of the same emotion differed only for anger

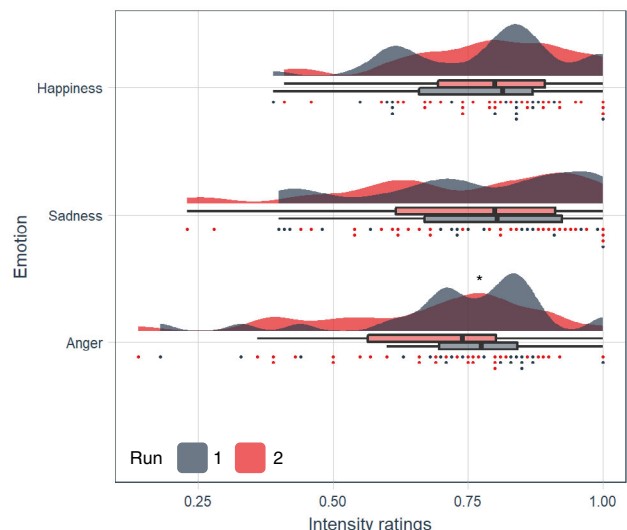

**Fig. 1 Combined raincloud-box and whisker plots of the intensity ratings on the scale of 0–1.** The subjects plotted as individual datapoints for each emotional event. The box bounds the IQR (interquartile range) divided by the median, and Tukey-style whiskers extend to a maximum of 1.5 × IQR beyond the box. $n = 32$ healthy subjects. $*V = 321.5$, $p = 0.025$.

(anger: $p = 0.025$; sadness: $p = 0.368$: happiness: $p = 0.805$) (Fig. 1).

**General affect network (GAN).** The affect-sensitive areas in our dataset covered widespread occipito-temporo-parieto-fronto-insular and cerebellar cortices as well as subcortical structures including amygdala and thalamus. These regions show a large and widespread overlap with the GAN (Fig. 2).

**Between emotion category specificity.** We tested 8 model-specific predictions from BET and PCT (Table 1). In order to test H1 (each emotion category activates dedicated structures compared to any other category (BET)), each emotion category was contrasted with the each of the other categories. There were no significant results for any emotion category at voxel-wise whole brain level, indicating no emotion-category specific activation. At ROI (region of interest)-level, Shapiro-Wilk tests revealed that not all variables were normally distributed (all $p$'s > 0.008), so non-parametric ROI-analyses were performed. Of note, for the ACC and OFC, we used the entire region as a ROIs, as well as a sphere surrounding the peak of category-specific activations from previous studies. Wilcoxon signed rank exact tests revealed increased activity only for happiness vs anger in the entire ACC ($V = 489$, $p = 0.007$) (Fig. 3c), but not for happiness vs sadness or happiness vs neutral (all $V$'s < 423, all $p$'s > 0.083). The happy-responsive ACC cluster was not more active during the experience of happiness compared to each of the other categories (all $V$'s < 437, all $p$'s > 0.053). Furthermore, none of the OFC nor MPFC ROIs showed any significantly increased activation for their associated emotion category compared to any of the other categories (all $V$'s < 390, all $p$'s > 0.193) (Fig. 3a, b). Both whole brain and ROI level results were thus in conflict with the prediction of BET relating to dedicated emotion category-specific neural circuits.

**Within emotion category consistency.** Within emotion category between event conjunction analyses were performed to test H2 (different events of a single emotion category activate similar structures (BET)). This revealed no significant results for anger

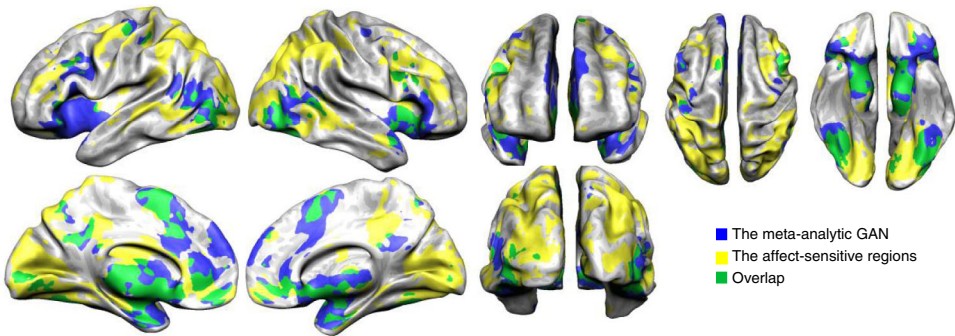

■ The meta-analytic GAN
■ The affect-sensitive regions
■ Overlap

**Fig. 2 The comparison of the affect-sensitive regions in our study to the meta-analytic general affect network (GAN).** The meta-analytic GAN, the affect-sensitive regions, and their overlap are shown on an inflated folded cortex.

| Table 1 Specific hypotheses derived from emotion theories. | | |
|---|---|---|
| | **BET** | **PCT** |
| Between emotion category specificity: | | |
| -H1: Each emotion category activates dedicated structures compared to any other category. | Yes | No |
| Within emotion category consistency: | | |
| -H2: Different events of a single emotion category activate similar structures. | Yes | No |
| -H3: The similarity between neural patterns across events within an emotion category (vs neutral) is significantly higher than between emotion category (vs neutral). | Yes | No |
| -H4: There is a significant overlap in regional activation across subjects within an emotion category. | Yes | No |
| -H5: The similarity between neural patterns across subjects within an emotion category is significantly higher than between emotion category. | Yes | No |
| Across emotion category consistency | | |
| -H6: The overlap in activation between emotion categories is high. | No | Yes |
| -H7: There is a significant association between emotion categories across subjects. | No | Yes |
| -H8: The similarity between neural patterns across events within an emotion category (vs baseline) is significantly higher than between emotion categories (vs baseline). | Yes | No |
| *BET* basic emotion theory, *PCT* psychological construction theory. | | |

and sadness, conflicting with the prediction of BET relating to consistency of different events of a single emotion category. For happiness, significant clusters were located in bilateral occipital pole, cerebellar vermis, and mesencephalon (Fig. 4), with only a limited overlap with the GAN. This result shows a neural consistency between different events of happiness as in line with BET, however the topography challenges BET since the consistent activations were mainly outside of the affect sensitive areas. H3 (the similarity between neural patterns across events within an emotion category (vs neutral) is significantly higher than between emotion category (vs neutral) (BET)) was tested by comparing correlations across events within an emotion category with correlations across events between different emotion categories. Shapiro-Wilk tests revealed that all pairwise between event correlations were normally distributed (all $p$'s > 0.103). One-tailed paired t-tests on the Fisher Z-transformed within category between event and between category between event Pearson correlation coefficients per emotion did not reveal any significant results in the GAN for any of the emotion categories (i.e., the between event within category correlations were not significantly stronger than the between event between category correlations for any of the emotion categories) (all $t(31)$'s < 1.635, all $p$'s > 0.056) (Fig. 5a–c), incompatible with BET. Furthermore, Shapiro-Wilk tests revealed that both all combined within category correlations and all combined between category correlations were normally distributed (all $p$'s > 0.374) and F test revealed that there was no significant difference on the variances of both groups ($p = 0.548$). Following one-tailed two-sample t-test on all combined Fisher Z-transformed within category between event correlations and all combined between category between event Pearson correlations

did not reveal any significant results in the GAN ($t(286) = 0.986$, $p = 0.162$) (Fig. 5d). These results conflict with BET prediction, showing lack of neural pattern consistency across events of an emotion category.

H4 (there is a significant overlap in regional activation across subjects within an emotion category (BET)) was tested by calculating the maximal percentage of subjects to obtain a minimal overlap in emotion category specific activation. First, we defined emotion-specific activation at a liberal threshold ($p < 0.05$, uncorrected) at subject-level. The resulting statistical map was binarized and probability maps across subjects were then computed. These revealed that each of 7 subjects (19%) showed activation in a cluster located in the culmen of the left cerebellum during anger experience (Supplementary Fig. 1), 6 subjects (16%) each activated two clusters located in right thalamus and the decline of right cerebellum during experience of sadness (Supplementary Fig. 2), and each of 7 subjects (19%) activated a cluster located in left primary visual cortex during the experience of happiness (Supplementary Fig. 3). The limited overlap (<20% for each emotion category) and the topography challenge the universal characteristics of basic emotions and the predefined brain-emotion associations by BET. Furthermore, H5 (the similarity between neural patterns across subjects within an emotion category is significantly higher than between emotion category (BET)) was tested by comparing correlations across subjects within an emotion category with correlations across subjects between emotion category. Similarity analyses revealed a heterogeneous pattern (Fig. 6a). Interestingly, the dissimilarity matrix clearly shows decreased dissimilarity within subject between emotion categories, compared to both between subject

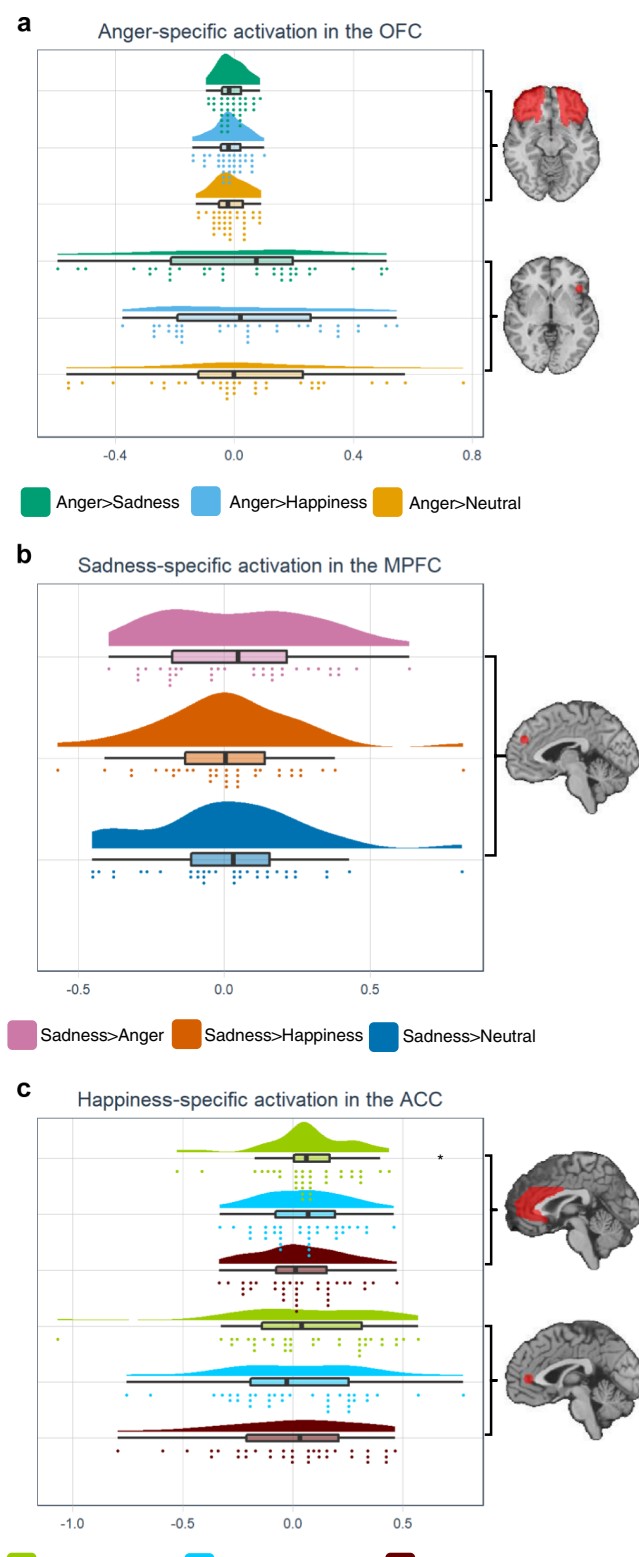

**Fig. 3 Emotion category specific activations. a–c** Combined raincloud-box and whisker plots of the beta values for anger-, sadness-, and happiness-specific activation in the OFC, MPFC, and ACC, respectively. The box bounds the IQR (interquartile range) divided by the median, and Tukey-style whiskers extend to a maximum of 1.5 × IQR beyond the box. n = 36 healthy subjects. *V = 489, p = 0.007. OFC orbitofrontal cortex, MPFC medial prefrontal cortex, ACC anterior cingulate cortex.

within emotion category and between subject between emotion category. The former reflects idiosyncratic across category neural patterns, in line with PCT. Shapiro-Wilk tests revealed that all pairwise between subject correlations were normally distributed (all $p$'s > 0.154), so parametric testing performed. F tests revealed there was a difference in variance between sadness-sadness vs sadness-happiness ($p = 0.012$). One-tailed Welch two sample t-tests revealed significant results for anger-anger vs anger-sadness ($t(1196.4) = 2.138$, $p = 0.016$), happiness-happiness vs anger-happiness ($t(1240.9) = 5.664$, $p < 0.001$), and happiness-happiness vs sadness-happiness ($t(1213.4) = 6.16$, $p < 0.001$) (Fig. 6b). However, the results did not reveal any significant result for anger-anger vs anger-happiness and for sadness-sadness vs any other combination (all $t$'s < 1.241, all $p$'s > 0.107). In addition, Shapiro-Wilk tests revealed that both pooled within category across subject correlations and pooled between category across subject correlations were not normally distributed (all $p$'s < 0.01). One-tailed Wilcoxon signed rank test showed significant results for pooled within category across subject correlations vs pooled between category across subject correlations ($W = 3781486$, $p < 0.001$) (Fig. 6c). These results show consistency between neural patterns across subjects for happiness and pooled across within emotion category correlations, in line with BET, however not for the categories anger and sadness.

**Across emotion category consistency**. Six pairwise between emotion conjunction analyses were performed to test H6 (the overlap in activation between emotion categories is high (PCT)). This revealed extensive overlap across a large portion of the brain (Fig. 7). H7 (there is a significant association between emotion categories across subjects (PCT)) was tested by performing between emotion category across subject correlation analyses for each of the 6 pairwise emotion category combinations. The resulting probability map revealed widespread significant results, revealing consistent inter-individual activity associations between emotion categories over large portions of the brain (Fig. 8). H8 (the similarity between neural patterns across events within an emotion category (vs baseline) is significantly higher than between emotion categories (vs baseline) (BET)) was tested using similarity analyses. Shapiro-Wilk tests revealed that not all pairwise between emotion correlations were normally distributed (all $p$'s > 0.009), so non-parametric analyses were performed. One-tailed paired Wilcoxon signed rank exact tests on the Fisher Z-transformed within category between event and between category between event Pearson correlation coefficients showed significant results for anger-anger vs anger-sadness ($V = 388$, $p = 0.010$) and anger-anger vs anger-happiness ($V = 375$, $p = 0.019$) in the GAN (Fig. 9a). However, the results did not reveal any other significant difference between within emotion combinations and between emotion combinations (all $V$'s < 304, $p$'s > 0.238) (Fig. 9b, c). Furthermore, Shapiro-Wilk tests revealed that both all combined within category correlations ($p = 0.031$) and all combined between category correlations ($p < 0.001$) were not normally distributed. One-tailed Wilcoxon signed rank test on all combined Fisher Z-transformed within emotion category between events and all combined between emotion category between events Pearson correlations did not revealed any significant result in the GAN ($W = 9585$, $p = 0.290$) (Fig. 9d). The overall results are more compatible with PCT instead of BET.

**Discussion**
The main objective of this study was to reveal fundamental neurobiological mechanisms associated with the experience of

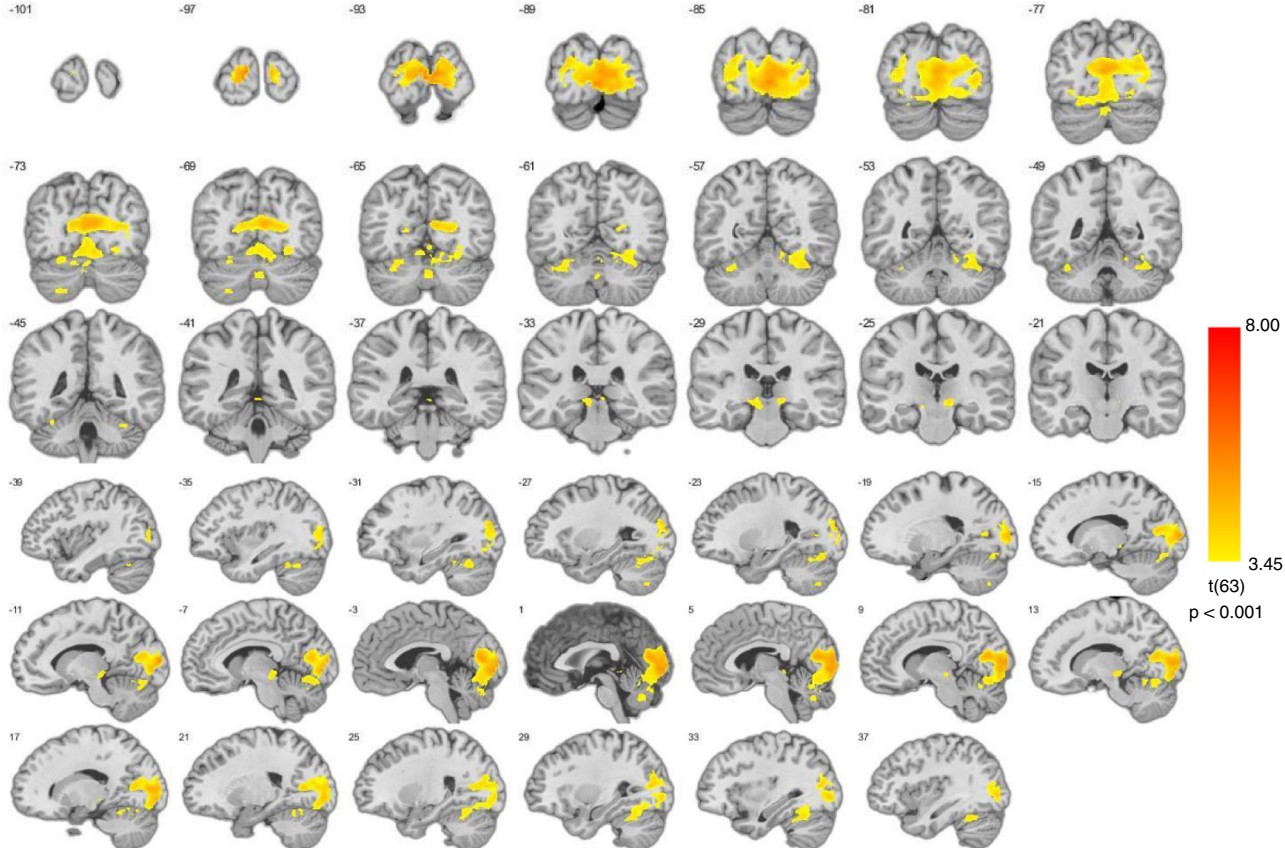

**Fig. 4 Regional overlap in activation between happy events.** Statistical map representing result of conjunction analysis between happy events presented on coronal and sagittal slices ($p < 0.001$).

emotion. We tested 8 predictions derived from conflicting emotion theories, i.e. BET and PCT. First, a GAN was defined based on independent meta-analytic findings[20] and the specificities and consistencies of neural amplitudes and patterns during experience of emotional events and categories (anger, sadness, and happiness) were investigated across the whole brain and within the GAN. The affect-sensitive regions in the present dataset showed significant overlap with the GAN in structures associated with emotion processing, such as anterior insula, amygdala, orbitofrontal cortex and thalamus. Subjective intensity ratings averaged over events did not significantly differ between emotion categories. Overall, the ratings were mainly high, and the values were similar for each emotion category. Subjective intensity ratings were statistically controlled for in the neuroimaging analyses.

Predictions relating to between emotion category specificity were tested on the regional activation amplitude (H1). We observed no emotion category-specific activation for any of the emotion categories, neither in the GAN nor in the according to BET a priori-defined OFC, MPFC, and ACC. The absence of emotion category specific-activation is in line with limited category-specific findings in emotion perception studies and meta-analyses[19,32] and conflict with BET predictions. Furthermore, the results of the ROI analyses in the OFC, MPFC, and ACC did not support the specific BET predictions of one-to-one structure to function mapping for anger, sadness, and happiness, respectively.

Within emotion category between event conjunction analyses (H2) again revealed no significant results for 2 of the 3 emotion categories. This strongly conflicts with the BET prediction that each instance of a particular emotion category activates its dedicated neural circuit. Instead, our findings indicate that multiple events of a single emotion show highly variable activation profiles.

However, for happiness, both events activated particularly the early visual cortices. This may be related to a vivid visual imagery of the happy events[43]. While overlapping activations for multiple events of a single emotion category are in line with BET predictions, the present topography is less so. Indeed, BET accounts associate the neural basis of happiness with the ACC[19], where we did not observe any overlap between happy events. In fact, only a limited part of the overlap clusters falls within the GAN or the meta-analytic map for positive affect[20]. Furthermore, predictions relating to within emotion category consistency were tested on the neural patterns. Between event similarity analyses (H3) revealed that pairwise correlations of different events within emotion category were significantly higher than the pairwise correlations of events of different emotion categories in the GAN, neither for any of the 3 emotion categories individually nor for all 3 emotion categories combined. This conflicts with one of the key predictions of BET on the consistency in activation patterns across events of a single emotion category while the findings can be explained by specificity and idiosyncrasy of each event in PCT.

Next, within emotion consistency was tested across subjects in regional activation and neural patterns. Probability maps of subject overlap in binarized regional within emotion category activation maps (H4) revealed very limited spatial overlap between subjects (<20% of all subjects) within each emotion category. Furthermore, predictions relating to within emotion neural pattern consistency across subjects (H5) were tested using similarity analyses. The results revealed that for 1 of the 3 categories (i.e. happiness) and for all categories combined, pairwise correlations across subjects within emotion were significantly higher than for any between emotion category combination. The finding that 2 out of 3 categories did not show this effect, conflicts

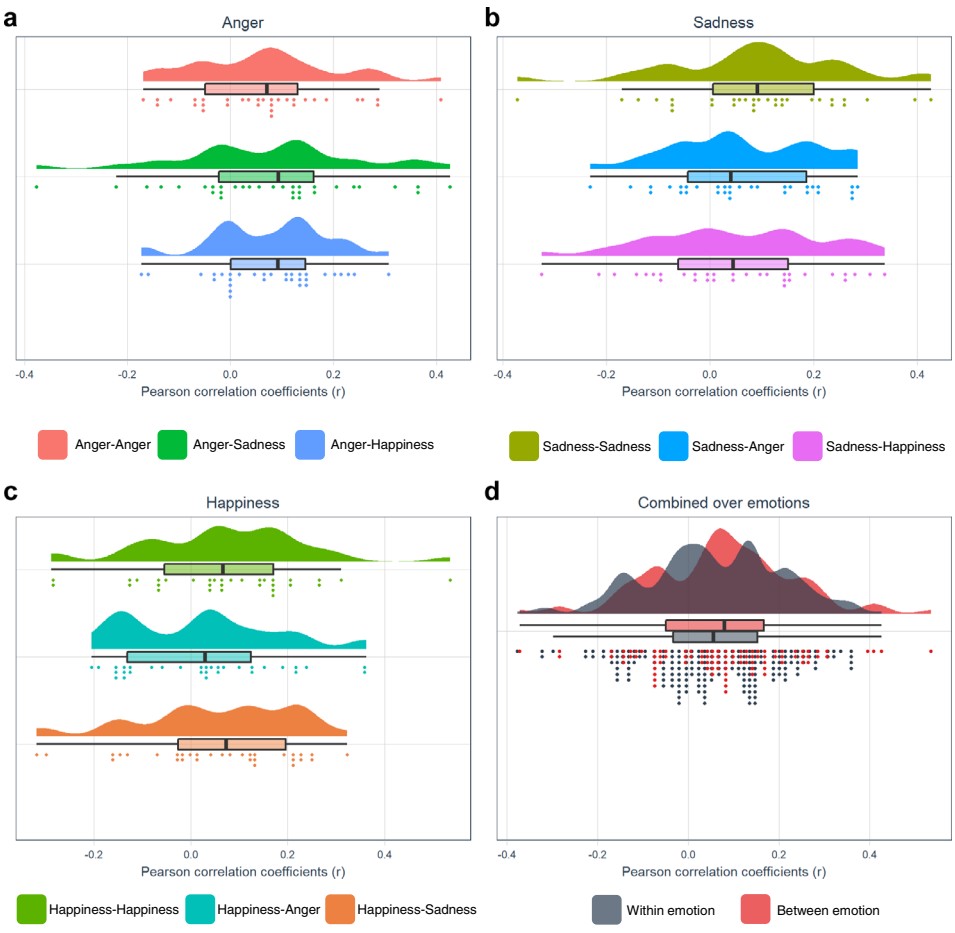

**Fig. 5 Combined raincloud-box and whisker plots of between event Pearson correlation coefficients (*r*) in the GAN. a–d** Anger vs neutral, sadness vs neutral, happiness vs neutral, and all emotions combined, respectively. The box bounds the IQR (interquartile range) divided by the median, and Tukey-style whiskers extend to a maximum of 1.5 × IQR beyond the box. *n* = 32 healthy subjects.

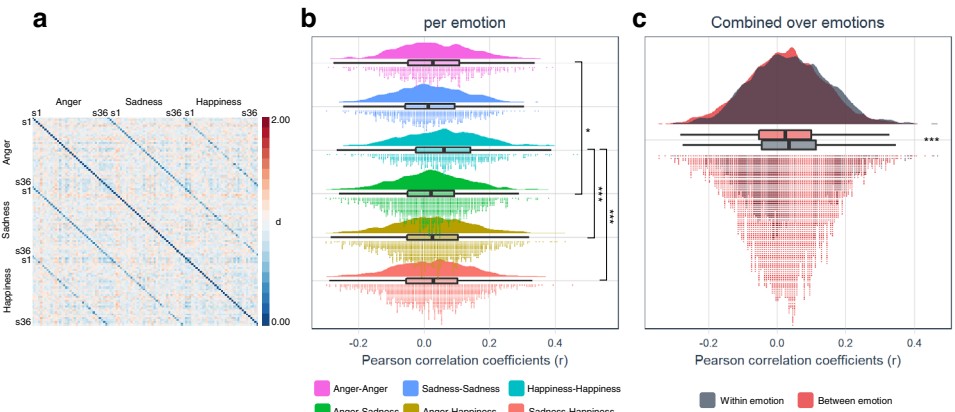

**Fig. 6 Within emotion category consistency across subjects. a** The dissimilarity matrix of the GAN, representing the neural pattern dissimilarities across subjects within and between emotion category. The dissimilarity matrix was generated using the distance metric (*d*) which is calculated by the equation '*d* = 1-*r*' where *r* is Pearson correlation coefficient, thus *d* values range from 0.0 (minimum distance, blue) to 2.0 (maximum distance, red) with 1.0 (no correlation, white) in the middle. 's' represents the subject numbers. **b, c** Combined raincloud-box and whisker plots of between subject Pearson correlation coefficients (*r*) in the GAN, per emotion and all emotions combined, respectively. The box bounds the IQR (interquartile range) divided by the median, and Tukey-style whiskers extend to a maximum of 1.5 × IQR beyond the box. *n* = 36 healthy subjects. *\*t*(1196.4) = 2.1384, *p* = 0.016; *\*\*\*p* < 0.001.

with BET, positing strong across subject consistency, adhering to the assumption of a genetic basis and universality.

Between emotion category conjunction (H6) and correlation (H7) analyses revealed extensive and widespread overlap. While this may partly result from a baseline effect, reflecting a task-general effect of re-experiencing an autobiographical event, it also reveals shared mechanisms across emotion categories. This result supports PCT, positing a common neural basis for all emotional

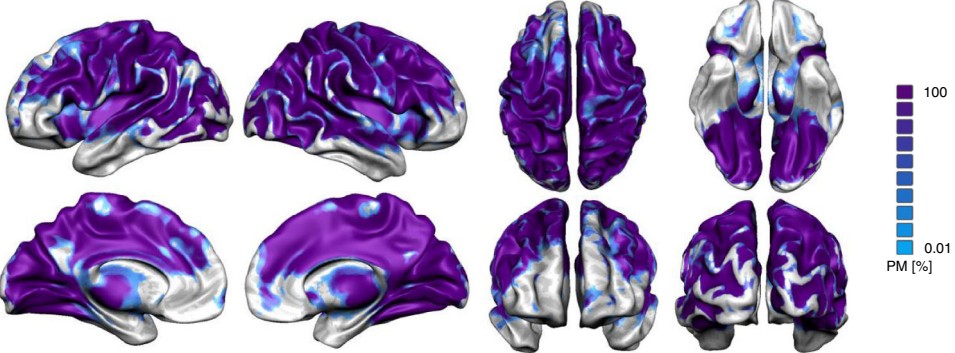

**Fig. 7 The overlap in activation between emotion categories.** Probabilistic map of spatial overlap of 6 pairwise between emotion category conjunction results represented on an inflated cortex.

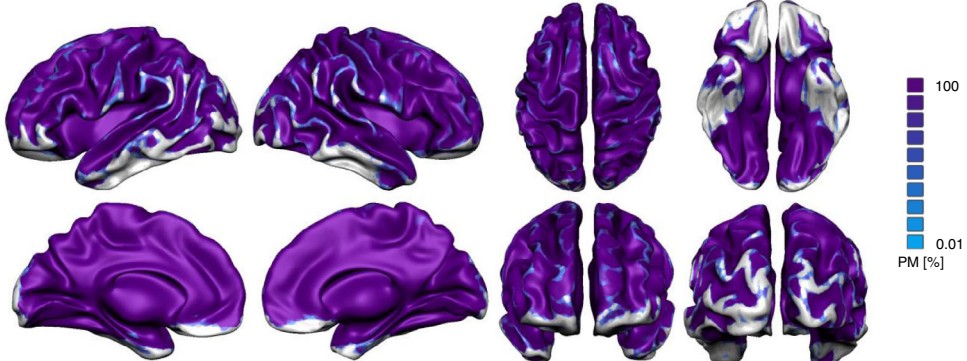

**Fig. 8 The association between emotion categories across subjects.** Probabilistic map of spatial overlap of 6 pairwise between emotion category correlation results represented on an inflated cortex.

(and also non-emotional) events, consisting of large-scale brain networks. Indeed, the conjunction results include the regions associated with default mode and salience network, as well as the semantic appraisal network. The between emotion category across subject correlations further suggest trans-categorical emotion traits. Subjects showing stronger activation compared to other subjects in e.g. the insula during experience of e.g. anger, will also show stronger activation compared to other subjects in the insula during experience of sadness. From the PCT perspective, it supports individual consistency in neural response across emotion categories, adhering to idiosyncratic mechanisms, compatible with the notion of 'neural topography trait'[44].

Finally, across category consistency was tested at the neural pattern level (H8). Pattern similarity analyses between events only revealed significant difference in the pairwise correlations of within emotion category vs between emotion category for anger in the GAN but not for any other pairwise correlations per emotion or over all emotions combined. This result shows that the similarity between neural patterns across events within an emotion category is not significantly higher than between emotion category for two of the three emotions. This result puts the within category between event association in perspective, as it indicates that the significant between event within category association is not limited to within category conditions, but extends across categories.

Several limitations of the present study need to be addressed. First, although the instructions to select emotional autobiographical events were standardized, neither specific suggestions nor restrictions regarding the type of events were provided. Nevertheless, there was some consistency in the event topics selected by the participants. For instance, typical examples for sad and happy events related to the passing and birth of loved ones, respectively, while anger events were related to arguments or disagreements with other people or unfair situations and neutral events were work, daily routines or chores. However, these content types were not systematic across the entire sample and hence the topics showed some variability across participants. This stands in strong contrast to conventional emotion perception studies, which typically use a single stimulus set for all participants. However, the present study focussed on standardizing the affective response in the participant, which is uncontrolled in mainstream emotion perception studies. Next, the subjects were instructed to select autobiographical events relating to intense emotion-category specific experiences. Following the scanning session, the majority of the participants indicated in the post-scanning interviews that they re-experienced the related emotions at high intensity. However, we do not have an additional control for neither the intensity of the original event nor whether the events were specifically related to a single emotion category. Although the subjects did not report that their autobiographical memories include more than one emotion, feeling multiple emotions in a single emotion block could bias the results favoring PCT. Furthermore, the strategy of re-experiencing emotional events for more than 1 min introduces susceptibility to a fluctuating intensity of affect within the event block and distraction related noise influences. However, it also allows to accommodate both steep and gradually increasing affect time-courses and to assume a reliable affect-specific average over the whole event duration. To assess the latter, we used a subjective index by means of self-rating. Next, the independent meta-analysis we used in order to define the GAN includes emotion experience studies as well as emotion perception studies. Yet there are significant

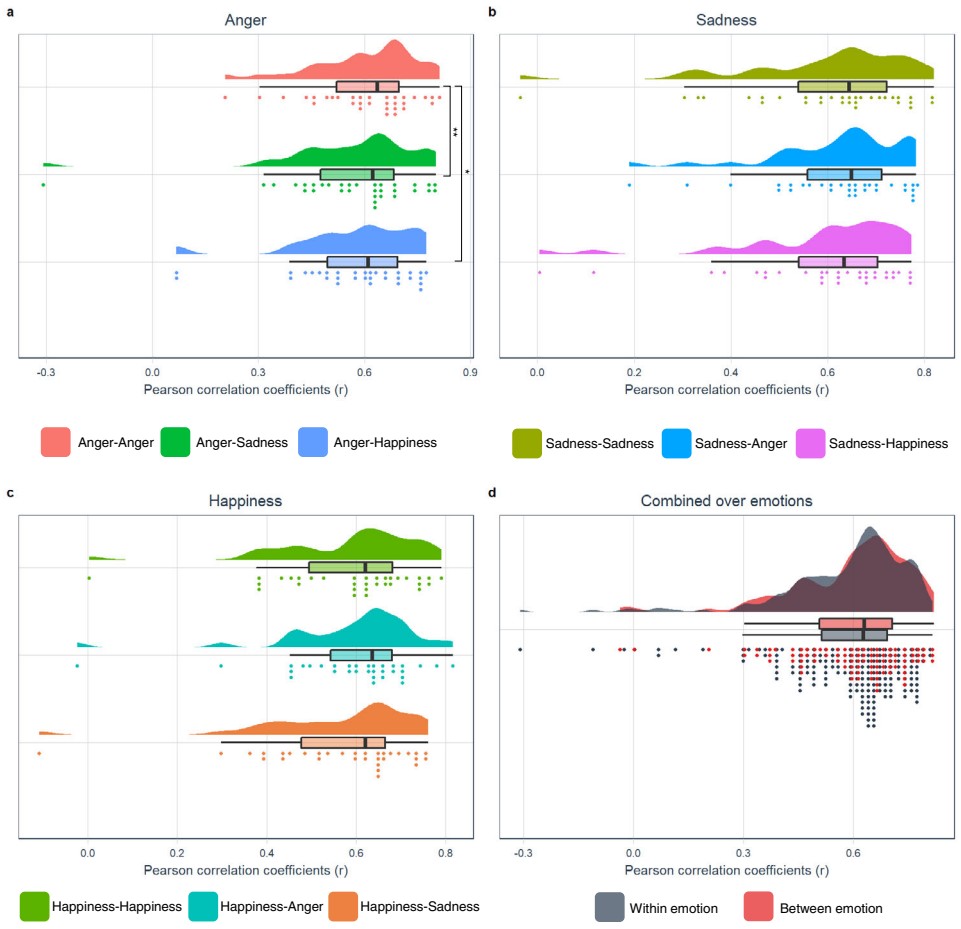

**Fig. 9 Combined raincloud-box and whisker plots of between event Pearson correlation coefficients (*r*) in the GAN.** **a–d** Anger vs baseline, sadness vs baseline, happiness vs baseline, and all emotions combined, respectively. The box bounds the IQR (interquartile range) divided by the median, and Tukey-style whiskers extend to a maximum of 1.5 × IQR beyond the box. $n = 32$ healthy subjects. $^{*}V = 375$, $p = 0.019$; $^{**}V = 388$, $p = 0.010$.

qualitative differences between the perception and experience of emotions. Emotion perception essentially reflects sensory (typically visual) processing of the external environment, which is often objectively standardized across participants as they are all shown the same stimuli. Emotion experience is a typically subjective process that has long been associated with processing of bodily sensations[45]. The basis of the GAN, including both perception and experience studies, does thus not constitute the ideal one for the present purposes as it may over- and underemphasize perceptual and experiential regions respectively. Indeed, comparing the affect-sensitive regions defined in our dataset with the GAN, reveals that some regions, e.g. the somatosensory cortices, show emotional modulation in the present dataset, but not in the meta-analytic map. Remarkably, we also observed emotional modulation in early visual regions outside the meta-analytic map. Of note, the participants had their eyes closed during the emotion experience event. We presume this may be explained by visual imagery effects[46]. The comparison between our affect-sensitive map and the meta-analytic map reveals that there are relevant affect-sensitive regions outside the GAN, despite the liberal threshold we applied to the GAN. However, as we opted for an independently defined GAN, we considered this one to be the most appropriate database currently available, as experience studies are uncommon. Furthermore, the GAN covers the key results of seminal emotion experience studies[33]. Further, the results may partly be influenced by imagery characteristics such as vividness. For the purpose of limiting task load and restricting scan duration, we did not add an imagery vividness rating to the

emotional intensity rating. Finally, the contrast of interest was adjusted as emotion conditions were compared to either the neutral condition or the implicit baseline for BET and PCT respectively. This was in line with the account for the claim of PCT of that similar mechanisms support both neutral and emotional events. Contrasting emotional with neutral condition would then filter out any processes of interest including conceptualization, language, and executive attention serving to construct emotional events[19].

We investigated within and between emotion category specificities and consistencies focussed on the neural basis of affect. Future studies may address these topics in other psychological modalities such as emotion regulation as well as other neural modalities like connectivity. Finally, it would be interesting to investigate neural changes in clinical populations with affective disorders and/or pathology in the GAN[47,48].

In conclusion, we tested 8 predictions from BET and PCT on the neural basis of affect and observed low between emotion category specificity and within emotion category consistency, conflicting with BET assumptions of biological inheritance and universality of emotions. On the other hand, the results revealed strong across emotion category consistency, compatible with PCT predictions on biological primitives underlying multifarious emotional states supported by large-scale networks.

## Methods

**Participants**. The study was approved by the Ethical Committee of University Hospitals Leuven (ML8040) and written informed consent of the participants was

obtained according to the Declaration of Helsinki. 37 healthy participants (19 females, mean age = 37 years, SD age = 12.656 years) were recruited via our database and public advertisements. All participants received monetary compensation.

**Experimental stimuli and paradigm**. At least two weeks prior to the brain imaging session, participants were instructed to select six autobiographical events that were associated with intense emotional experiences: two angry, two happy, and two sad events. Furthermore, they were instructed to think of two emotionally neutral autobiographical events. For each event, they were asked to provide a single word that would unambiguously be associated with the specific event.

The imaging experiment consisted of two functional runs, each containing one event of each emotion category. A run started with a 1000 ms presentation of a black screen, followed by presentation of one of the provided words for 3000 ms. Subsequently, "Close your eyes now" was displayed for 2000 ms. During the following 61 s, a black screen was presented, which constituted the emotion experience block. Prior to the scanning, participants were instructed to try to re-experience the respective emotion during this block as intensely as possible. The end of this block was signalled to the participant by three alternating 500 ms presentations of black and white screens, which were easily detectable with the eyes closed. Subsequently, an emotion intensity rating was performed by means of a visual analogue scale. This event consisted of a 10 s presentation of a slider on which participants could rate how intensely they had re-experienced the emotion (from 'very weak' to 'very intense'). After this interval, the text "Press after vertical line" was displayed for 4000 ms. Subsequently, 30 stimuli consisting of a circle filled with line gratings were randomly presented one by one for 500 ms each with a 500 ms inter-stimulus interval. Five stimuli displayed vertical gratings and 25 displayed horizontal gratings. This cognitively demanding visual reaction time task was included to minimize emotional carry-over effects between emotion categories. Subsequently, this procedure was repeated 3 times within a run, albeit each with a different provided word presented. The order of emotion categories was counterbalanced. For two of the participants, the duration of the experimental procedure was slightly different due to 1000 ms instead of 500 ms presentations of the line grating stimuli. A high-resolution structural scan was performed in between both functional runs (see also[49]).

**Image acquisition**. The functional magnetic resonance imaging (fMRI) was performed using a 3 Tesla (MR) scanner (Achieva3T; Philips, Best, the Netherlands) with a 32-channel head coil. Functional runs had a duration of 507 s each (532 s for the two participants with a slightly different protocol). 140 T2*-weighted Blood-Oxygenation Level Dependent (BOLD) contrast volumes were acquired (153 and 150 volumes for the two deviant participants). A functional volume consisted of 70 axial slices oriented parallel to the anterior commissure—posterior commissure (AC-PC) plane with 2.0 mm slice thickness, no gap, 2.75 × 2.75 mm in-plane resolution, 80 × 80 matrix size, and 220 × 220 mm field of view (FOV) and covered the whole brain. The echo time (TE) (26 ms), the repetition time (TR) (3500 ms) and flip angle (90°) were optimized for subcortical sensitivity[50]. The first four volumes were dummy volumes to allow for T1 equilibration. In between the functional runs, a high resolution T1-weighted anatomical image with 1 × 1 × 1 mm voxel size was acquired by using a three-dimensional (3-D) magnetization-prepared-rapid acquisition gradient echo sequence as 182 slices with 4.6 ms of TE, 9.6 ms of TR, 256 × 256 matrix size.

**Data analysis**. First, the intensity ratings are calculated on a scale from 0 to 1 depending on the position of the slider (visual analogous scale) as from 'very weak' to 'very intense'. Then, we tested whether the intensity ratings averaged over events were normally distributed using Shapiro-Wilk tests. Depending on the resulting normality, the differences in intensity ratings between emotion categories were evaluated by means of parametric or non-parametric testing. Furthermore, we tested whether the intensity ratings across subjects per event were normally distributed using Shapiro-Wilk tests. Then again depending on the normality, we tested whether the intensity ratings of two events of the same emotion category differed using parametric or non-parametric paired testing.

The whole brain voxel-wise statistical threshold was set at $p < 0.05$ FDR-corrected combined with a maximal uncorrected $p$-value of 0.001. Minimal cluster size was established via 1000 Monte Carlo simulations of random image generation, followed by the injection of spatial correlations between neighboring voxels, voxel intensity thresholding, and cluster identification.

**Pre-processing**. Pre-processing included realignment for motion correction, slice time correction, coregistration of the anatomical and functional images, spatial normalization of both anatomical and functional images to Montreal Neurological Institute (MNI) standard space with 2 × 2 × 2 mm voxel size, and smoothing of the images using a Gaussian kernel of 8 mm full width at half maximum (FWHM) as final step. Both runs of one participant and one run of three participants were excluded due to excessive motion in the scanner (>3 mm). One run of one participant was excluded due to inaccurate understanding of the instruction. This resulted in 36 participants with at least 1 run and 32 participants of which both runs were included.

**Random effect general linear model**. A first level general linear model (GLM) was constructed with 9 regressors defined by the onset and offset of the 4 emotion experience epochs, the 4 visual reaction time epochs, and a constant, convolved with the canonical hemodynamic response function. The entire 61 s of emotion experience event was modelled as single event and the intensity ratings given by the subjects were included as weights in the GLM. Next, a random effects (RFX) group GLM analysis was performed using a second-order autoregressive model for removal of serial correlations and time course normalization using z transformation.

**Region of interest (ROI) definition**. First, we used an independent dataset to define a general affect network (GAN). This was based on the results of a meta-analysis of 397 neuroimaging studies of emotion experience and perception[20]. The meta-analysis tested 3 hypotheses regarding the neural basis of emotional valence, i.e., positive and negative affect, and revealed a flexible set of valence-general limbic and paralimbic structures[20]. In particular, the GAN was defined by combining the meta-analytic maps for negative and positive affect. We used an inclusive approach and defined the GAN as the pooled clusters of each map, using a lenient threshold of 0.06 on each proportion map. In addition, we combined a global and local approach in addressing a priori associations between emotion category and brain region according to BET. The global approach consisted of anatomical definition the entire OFC and ACC using the 'automated anatomical labelling' atlas[51] and the local approach of defining 6 mm radius spherical ROIs centred around peak coordinates that show consistent responses to anger, sadness, and happiness, based on meta-analytic findings respectively in the OFC, MPFC and ACC[15].

**Comparison affective responses with meta-analytic findings**. We defined affect-sensitive regions in our dataset by means of a voxel-wise whole brain single factor analysis of variance (ANOVA) with four levels, corresponding to the four emotion category conditions, and compared the resulting regions to the GAN defined on a meta-analytic basis. To maximize the similarity between the meta-analytic map and the present results, the intensity ratings were not included as regressor weights in the definition of the affect-sensitive regions.

**Between emotion category specificity**. Emotion-specific activation was investigated by contrasting each emotion category with each of the 3 other categories via voxel-wise whole brain conjunction analyses, e.g. for anger: anger>sad ∩ anger>happy ∩ anger>neutral. Furthermore, we calculated the beta-values (averaged over voxels) for anger vs each of the other categories, sadness vs each of the other categories, and happiness vs each of the other categories in the OFC, MPFC, and ACC ROIs, respectively. First, the normality of the distribution of beta-values was tested by means of Shapiro-Wilk tests and depending on the result, tested against zero by means of parametric or non-parametric testing.

**Within emotion category consistency**. Regional overlap in activation between events of a single emotion category was investigated by means of voxel-wise whole-brain within emotion category between events conjunction analyses. We contrasted each emotion category with the within run neutral condition and subsequently ran 3 conjunction analyses, one for each of the 3 contrasts (e.g. conjunction of anger vs neutral in the first run with anger vs neutral in the second run), resulting in three statistical overlap maps, one for each emotion category. Furthermore, to investigate the similarity of the neural patterns within emotion category across events, we performed similarity analyses between 1 of the 2 events of each emotion category and the remaining events of each emotion category (e.g. similarity analyses between anger vs neutral in the first run and anger vs neutral in the second run, between anger vs neutral in the first run and sadness vs neutral in the second run) in the GAN. In order to investigate whether the pattern similarity across events within an emotion category is higher than between emotion category, for each emotion category, pairwise Fisher Z-transformed Pearson correlation coefficients between events within emotion category were compared to pairwise Fisher Z-transformed Pearson correlation coefficient between events of that emotion category and events of other emotion categories (e.g. whether the correlations between anger vs neutral in the first run and anger vs neutral in the second run is significantly higher than correlations between anger vs neutral in the first run and sadness vs neutral in the second run). First, the normality of the distribution of pairwise Fisher Z-transformed Pearson correlation coefficients between events was tested for each variable individually, pooled within emotion correlations, and pooled between emotion correlations by means of Shapiro-Wilk tests. Depending on the results, the assessment of the significance was performed by means of 2 parametric or non-parametric paired testing on sets of 32 within subject between event Fisher Z-transformed correlations for each emotion category separately and by parametric or non-parametric testing on pooled within emotion correlations on the one hand and pooled between emotion correlations on the other hand. Next, to investigate spatial overlap between subjects within emotion categories, we first identified for each subject the regions that were active during each emotion category vs neutral at a liberal threshold ($p < 0.05$, uncorrected), biasing the methods in favour of BET, and subsequently created binary maps, of which we calculated the percentage overlap across subjects for each emotion category. We then calculated the maximal percentage of subjects to obtain a minimal overlap. Furthermore, to investigate the

similarity of neural patterns within emotion categories across subjects, we performed similarity analyses across subjects for within each emotion category and between each emotion category pair, revealing all 630 (for each within emotion category) and 1260 (for each between emotion) pairwise between subject Pearson correlations of the neural patterns in the GAN[37]. In order to investigate whether the pattern similarity across subjects within an emotion category is higher than between emotion category, we compared pairwise Fisher Z-transformed Pearson correlation coefficients between subjects within emotion category to pairwise Fisher Z-transformed Pearson correlation coefficients between subjects between emotion category for each emotion category separately and all emotion categories combined (pooled within emotion categories vs pooled between emotion categories). First, the normality of the distribution of all pairwise Fisher Z- transformed Pearson correlation coefficients was tested using Shapiro-Wilk tests for each variable individually, pooled within emotion correlations, and pooled between emotion correlations. Depending on the normality, the assessment of pattern similarity significance was performed by means of 2 parametric or non-parametric paired testing on all pairwise Fisher Z-transformed Pearson correlation coefficients for every emotion category separately and by parametric or non-parametric testing on pooled within emotion correlations and pooled between emotion categories.

**Across emotion category consistency**. We performed 6 voxel-wise whole-brain conjunction analyses between activation for each of the six emotion category pairs (anger-sad; anger-happy; anger-neutral; sad-happy; sad-neutral; happy-neutral). Each of the emotion conditions was compared to implicit baseline and not to the neutral condition, as PCT posits that similar mechanisms support both neutral and emotional events, in line with recent findings on emotion processing[52]. We also investigated whether there were associations between emotion categories in the regional activation level across subjects by calculating voxel-wise whole-brain Pearson correlation coefficients between each of the six emotion category pairs (anger-sad; anger-happy; anger-neutral; sad-happy; sad-neutral; happy-neutral). In addition, similarity analyses between events were performed exactly in the same approach as the similarity analyses between events to test within emotion category consistency but emotion conditions were compared to implicit baseline instead of to the neutral condition. In order to investigate whether within emotion between event correlations are different than the between emotion category between event correlations, the assessment of the significance was performed again as explained above.

**Statistics and reproducibility**. Imaging data analysis was performed using BrainVoyager 22.0 (Brain Innovation, Maastricht, The Netherlands)[53], Neuroelf v1.1 (http://neuroelf.net) within MatLab R2020b (Mathworks, Inc) and statistical analyses were performed using RStudio Desktop 2022.07.1 + 554[54] within R 4.2.1[55]. All the statistical analyses except between event analyses ($n = 32$) were performed on the whole participant sample ($n = 36$) after the exclusion of 1 participant due to excessive motion in the scanner. The details about experimental design and statistics used in different analyses performed in this study are given in the respective sections of results and methods.

**Reporting summary**. Further information on research design is available in the Nature Portfolio Reporting Summary linked to this article.

## Data availability
The data that support the findings of this study are available from the corresponding author, upon request. Data are still being analyzed for other purposes and cannot be made publicly available at this time. The source data underlying the graphs are provided as Supplementary Data 1.

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

## Acknowledgements

This study was funded by KU Leuven Fund (C24/18/095 & IDN/21/010) and the Sequoia Fund for Research on Ageing and Mental Health. We are grateful to Ajay Satpute and Jiahe Zhang for sharing the neuroimaging files.

## Author contributions

D.G., R.P., S.S., L.V.O., M.V., L.F.B. and J.V.d.S. designed the research; F-L.D.W. and J.V.d.S. performed the research; D.G., J.P., A.E.K. D.S., S.S., L.E. and J.V.d.S. analyzed the data; D.G. and J.V.d.S. wrote the paper with input from all authors; D.G., J.P., L.V.O., M.V., L.F.B. and J.V.d.S. contributed ideas for statistical analyses and interpretation.

## Competing interests

The authors declare no competing interests.
