## [Peer Review File · Communications Biology]

Reviewers' comments:

Reviewer #1 (Remarks to the Author):

This paper examines the neurobiological basis of emotion using fMRI to evaluate whether neurological activity recorded during affective experience is more consistent with the classic Basic Emotions Theory (BET) or the more modern Psychological Construction Theory (PCT). Using an autobiographical memory recall task the authors elicited three different emotions (and a neutral experience). Results indicated that neural activation during emotional experiences was better predicted by the PCT than BET, with strong evidence for a common neural basis of affect, shared across different emotion categories. I think the paper addresses an interesting question and I like the authors original approach of eliciting emotions using memories rather than previous work that has for example evaluated processing of emotion expressions. The paper was generally clearly and concisely written, I do however have a couple of smaller comments.

There are a large number of hypotheses tested and while they obviously cannot be changed I feel it is possible they could be clarified and potentially made a little more concise. I found myself frequently having to refer back to them to make sense of the paper, some clarification or reiteration could make this unnecessary. In addition, some of the hypotheses seemed ill-defined (or not clearly operationalised) which made the interpretation of the results feel somewhat subjective. Small revisions of the discussion of the results could clarify explanations of why BET/PCT were/were not supported.

The definition of BET utilised when evaluating the results is quite stringent and the discussion around the results and whether they support BET specifically could be a little more nuanced. Indeed, BET researchers acknowledge that emotions arise from integrated neural circuitry e.g. "Emotion feelings arise from the integration of concurrent activity in brain structures and circuits that may involve the brain stem, amygdale, insula, anterior cingulate, and orbitofrontal cortices" (Izard, 2009, pg.5).

While I applaud the authors in their aim to maximise the similarity of experienced emotion/emotional response in participants (line 147) this seems to have only been evaluated by assessing intensity during the study. More detail here would be very helpful, if possible, for example was the intensity of the original event assessed? In addition were emotions specifically probed/assessed to ensure that memory recalled was specifically happy? It may be that memories have more than one emotion which would bias the results into being likely to support PCT. This limitation could be added to the discussion, and the limitation of the methodology discussed a little further. Additionally, the manuscript indicates that some information was collected about the autobiographical events (line 338), it would be helpful to have more information included in the article about these events to help understand the emotion experience of the participants.

For the across emotion category analysis the justification for using baseline rather than the neutral condition could be further discussed, as while it makes sense that PCT suggests similar neutral mechanisms will be utilised and therefore a baseline is used, the comparison with neutral would reduce noise associated specifically with the autobiographical recall required by the task (line 492).

The discussion could explore in a little more detail the meaning and interpretation of the neural activity related to happiness (H3).

Overall, I think this is an interesting paper examining the neural mechanisms underlying experienced emotions and I commend the authors on their original approach to this research question and wish them well in their future work.

Reviewer #2 (Remarks to the Author):

The study by Gundem et al., assesses (patterns of) BOLD activation related to the retrieval of distinct emotional autobiographical memories with the aim to test two competing hypotheses about how emotions are represented in the brain. The authors find that their result fit better with

Psychological Construction Theory (PCT) than with (stringent forms of) Basic Emotion Theory (BET). The paper is well-written, and the approach has potential. However, I am concerned that the main analyses and their interpretations are fundamentally flawed, with two confounds explaining why between-emotion-category consistency might seem higher than within-emotion-category consistency:

1. It seems that within-emotion between-event conjunction/classification/RSA (H1-6) is done between functional runs, while between-emotion conjunction/classification/RSA (H7-H9) is done within functional runs. Is this correct? That makes a direct comparison between the two (H10) problematic, as higher similarity/ classification/ activation overlap in the present results could merely reflect the temporal distance between the events (see e.g., Mumford, J. A., Davis, T., & Poldrack, R. A. (2014). The impact of study design on pattern estimation for single-trial multivariate pattern analysis. *Neuroimage*, 103, 130-138). The solution to this problem might be simple: between-run analyses only. This might not leave enough events for SVM, but RSA is sufficient to test the hypotheses.

2. The authors adjust the contrasts of interest based on the theories, which precludes direct comparisons of the theories. For within emotion category consistency, emotional memory retrieval is contrasted with neutral memory retrieval, properly control for non-emotional experiences (i.e., memory processing). Not much survives in the conjunction analyses and RSA values in the GAN are low, but significant. There could be a power issue, but the results do speak against stringent forms of BET. So far so good. But then for the across emotion category consistency emotional memory retrieval is contrasted with implicit baseline, which obviously reveals much higher correlations across emotions, because there is no control for the fact that participants are doing something cognitively demanding (so pattern similarity might be driven by this cognitive demand/ memory retrieval, nothing to do with emotions). It is not tested whether within-emotion category similarity explains any additional variance on top of the variance explained by between emotion-category. And this is necessary, because if the assumption is that the task is reliable, then a similar event should evoke similar BOLD patterns, at least as similar as the BOLD pattern related to a different event (would be best tested with Bayesian analyses). There is no theory that argues why between-emotions patterns should be higher than within-emotions between-events patterns, and it is unclear why the authors would have predicted this (H10).

3. Relatedly, the absolute correlation values in RSA analyses shouldn't be interpreted as 'high' or 'low' (even if point 1 & 2 are addressed). There are so many factors influencing these absolute values (e.g., filtering, power, specific task, specific contrast). fMRI is a noisy measure, and the design has very few events - in the end it is about whether there is any variance explained by within-emotion category similarity above and beyond between-emotion category. If so, that is evidence for (less stringent forms of) BET, if not, it supports PCT. But the bottom line is that the comparisons need to be done on the same contrast (either emo vs neutral, or emo vs baseline for both within and between emotion category similarity) and between functional runs only.

Minor

4. The autobiographical memory retrieval is used as an emotion induction. Although intensity does not significantly differ between emotional categories, variances didn't seem equal (figure S1). Is there a reason why intensity is not included in the model, as regressor of no interest?

5. If I understand it correctly, participants re-experienced their emotion for 61 seconds. It wasn't quite clear to me from the description how this was modeled in the GLM. Is the entire minute modeled as one event?

6. Regarding the data availability: the authors state that the data cannot be archived publicly because of confidentiality. I assume this regards the autobiographical memory scripts, and this makes sense. However, there is no reason why the other data could not be made available (i.e., affect labels of the events / ratings scales / raw, defaced imaging data). It is the sharing of this data that allows other researchers to reproduce results (data availability 'upon request' puts up unnecessary barriers to such attempts).

Reviewer #3 (Remarks to the Author):

Summary:

The manuscript describes an fMRI study investigating similarities vs. differences (across/between emotion categories, events and subjects) in neural responses during the re-experience of different emotions (happy, angry, sad vs neutral) elicited by autobiographical memory. The goal of the study was to investigate (and contrast) specific predictions of two influential emotion theories: basic emotion theory (BET) vs. psychological construction theory (PCT). The authors conclude that the results are more consistent with PCT than BET, given that the results provide stronger support for a common neural basis shared across categories (as predicted by PCT) than for category-specific neural patterns or consistency across different events within the same emotion category (as predicted by BET).

General impression:

The introduction and predictions are clear and follow logically from the two theories. The conclusions are also relatively clear, although there is some ambiguity in the discussion with regards to support for (less stringent versions of) BET (see minor comment 7 below). As I am not a neuroscientist, I cannot judge the fMRI methods and results in particular. I do have some more general questions and comments with regards to the analyses, results and conclusions. Please find my detailed comments below.

Major comments:

1. The dataset used for the study is relatively small, $n=36$ (or 32 for some analyses) and only 2 instances of 4 emotions each. I wondered if this dataset is large enough (has enough power) to detect differences between emotion categories, similarities between events within a category, and whether there are enough datapoints for reliable SVM classification (which is based on a split dataset (1 event per category each) for training and testing). More importantly, I wondered if a possible lack of power perhaps biases the findings against BET and in favour of PCT, as BET predictions seem to be based on analyses of (pairwise) differences between categories or similarities between events within a category (small subsets of the data), whereas PCT predictions seem to be based on analyses across categories (all data). Could the authors provide a power calculation for the different analyses and explain whether this may be a problem or not (i.e., is there a possible lack of power and if so, could this differentially affect the support for the two theories (or why not)?
2. The results state that intensity ratings did not differ between emotion categories, but there seems to be quite some variation within categories (figure S1), i.e., between subjects and events. I wondered to what extent this may have affected the conclusions of the study: If similar neural regions were activated, but with different intensity, how would that affect the different analyses (in particular conclusions with regards to (dis)similarity of neural patterns)? I think this is important as several analyses were based on within-category comparisons. Also, it would be helpful to see the variation between subjects and events plotted as individual datapoints in the figure (S1).

Minor comments:

3. It is not entirely clear to me, in the introduction, what the difference is between H3, H4, and H5. Don't they in fact test the same BET hypothesis (i.e., different events of the same category activate similar neural patterns), but perhaps only with different analysis methods?
4. It is not clear to me why, according to PCT, the neural consistency is stronger across than within categories (H10, introduction line 136-137, and discussion line 330-331). Given theorized emotion-general elicitation mechanisms (across categories) and individual specificity (within categories), I can understand why PCT would perhaps posit similar consistency across as within categories. But why would the neural patterns be more similar for events that elicit different emotion experiences (consistency across categories) than for events with a similar emotion experience? Please explain.
5. Intensity ratings: It is not clear from the methods (line 384) and results (line 163) on which scale the ratings were performed, which makes it difficult to interpret these findings. Please add the scale range in the description.
6. The meaning of the results is often not clear (for a reader unfamiliar with the specific analyses). It would be helpful if the hypotheses were briefly repeated here (instead of only referring to the

number, e.g., H1), and whether the result supported the hypothesis or not. E.g., line 177: "no significant results for any emotion category at voxel-wise whole brain level". Does this mean no similarities or no differences between categories? And is this in line with H1 or not? I find the following results similarly unclear (please explain what this means in reference to the respective hypothesis):

- Line 186: The trained classifier performed above chance, but only when taking the pattern in the GAN as a whole into account.
- Line 197: No significant results for anger or sadness.
- Line 197-198: significant clusters, but a limited part of these fell within the GAN.
- Line 200: The classification did not perform above chance
- Line 204 and 211: Significant yet low positive correlations
- Examples where I did find the results clear are e.g. line 231: "analyses revealed extensive overlap across the entire brain", or line 234: "revealing consistent inter-individual activity associations between emotion categories over large portions of the brain". Please explain the other findings in a similar fashion.

7. In the discussion, there is some ambiguity with regards to the extent to which some findings support BET or not, as there are more and less stringent formulations of BET. It seems that some findings do support some predictions of (less stringent versions of) BET:

- Line 276 (H2): "in line with predictions from less stringent variants of BET".
 - Line 288 (H3): "for happiness, both events activated the bilateral occipital pole, cerebellar vermis, and mesencephalon."
 - Line 300 (H4): "the patterns show some relevant consistency between events of a single category".
 - Line 303 (H6): "associations ranging between weakly negative and moderately positive".
- But the overall conclusions of the authors do not seem to really acknowledge that:
- Line 281: "Overall, these findings are incompatible with less stringent variants of BET".
 - Line 301: "the limited effect challenges essentialist BET predictions".
 - Line 306: "Findings are only mildly compatible with BET predictions".

I wondered how predominant stringent vs. less stringent versions of BET are in the contemporary literature. Aren't more modern versions less stringent? If that is so, I think that deserves acknowledgement in the discussion (e.g., results are not in line with original stringent BET, but do partially support (less stringent) modern versions).

8. The selection of ROIs was based on a meta-analysis that included both studies that investigated emotion perception and studies investigating emotion experience (see line 435-436). However, according to the authors there may be important differences in emotional responses with these methods (see line 79-80). I wondered if/how the inclusion of perception studies in the ROI selection, while the focus of the current study was on experience, may have influenced the results.

Response to Reviewers

We thank the Reviewers for their constructive comments and their praise of the manuscript and the study as “I think the paper addresses an interesting question and I like the authors original approach of eliciting emotions using memories rather than previous work that has for example evaluated processing of emotion expressions. The paper was generally clearly and concisely written.”, “Overall, I think this is an interesting paper examining the neural mechanisms underlying experienced emotions and I commend the authors on their original approach to this research question.” “The paper is well-written, and the approach has potential.”, “The introduction and predictions are clear and follow logically from the two theories. The conclusions are also relatively clear.”

Reviewer #1

Comment #1:

There are a large number of hypotheses tested and while they obviously cannot be changed I feel it is possible they could be clarified and potentially made a little more concise. I found myself frequently having to refer back to them to make sense of the paper, some clarification or reiteration could make this unnecessary. In addition, some of the hypotheses seemed ill-defined (or not clearly operationalised) which made the interpretation of the results feel somewhat subjective. Small revisions of the discussion of the results could clarify explanations of why BET/PCT were/were not supported.

Response #1:

We thank Reviewer #1 for pointing this out. We have followed Reviewer #1’s suggestions and clarified the hypotheses, reiterated them and discussed the interpretation of the results within the related sections more explicitly in the revised manuscript. In addition, in order to ease the understanding and follow up of the hypothesis, we also added a table (line 147: Table 1: Specific hypotheses derived from emotion theories.).

Comment #2:

The definition of BET utilised when evaluating the results is quite stringent and the discussion around the results and whether they support BET specifically could be a little more nuanced. Indeed, BET researchers acknowledge that emotions arise from integrated neural circuitry e.g. “Emotion feelings arise from the integration of concurrent activity in brain structures and circuits that may involve the brain stem, amygdala, insula, anterior cingulate, and orbitofrontal cortices” (Izard, 2009, pg.5).

Response #2:

We agree with Reviewer #1 and we have rephrased the sections on the evaluation of the results related to BET, and have included the mentioned reference by the Reviewer #1 as follows:
Line 47: “BET claims that emotions arise from integrated neural circuitry including the brain stem, amygdala, insula, anterior cingulate, and orbitofrontal cortices¹”

Comment #3:

While I applaud the authors in their aim to maximise the similarity of experienced emotion/ emotional response in participants (line 147) this seems to have only been evaluated by assessing intensity during the study. More detail here would be very helpful, if possible, for example was the intensity of the original event assessed? In addition were emotions specifically probed/assessed to ensure that memory recalled was specifically happy? It may be that memories have more than one emotion which would bias the results into being likely to support PCT. This limitation could be added to the discussion, and the limitation of the methodology discussed a little further. Additionally, the manuscript indicates that some information was collected about the autobiographical events (line 338), it would be helpful to have more information included in the article about these events to help understand the emotion experience of the participants.

Response #3:

We have followed Reviewer #1's suggestions, have provided additional information on autobiographical events and further discussed the control issues within the limitations in the revised manuscript as follows:

Line 328: "First, although the instructions to select emotional autobiographical events were standardized, neither specific suggestions nor restrictions regarding the type of events were provided. Nevertheless, there was some consistency in the event topics selected by the participants. For instance, typical examples for sad and happy events related to the passing and birth of loved ones, respectively, while anger events were related to arguments or disagreements with other people or unfair situations and neutral events were work, daily routines or chores. However, these content types were not systematic across the entire sample and hence the topics showed some variability across participants."

Line 338: "Further, the subjects were instructed to select autobiographical events relating to intense emotion-category specific experiences. Following the scanning session, the majority of the participants indicated in the post-scanning interviews that they re-experienced the related emotions at high intensity. However, we do not have an additional control for neither the intensity of the original event nor whether the events were specifically related to a single emotion category. Although the subjects did not report that their autobiographical memories include more than one emotion, feeling multiple emotions in a single emotion block could bias the results favoring PCT."

Comment #4:

For the across emotion category analysis the justification for using baseline rather than the neutral condition could be further discussed, as while it makes sense that PCT suggests similar neutral mechanisms will be utilised and therefore a baseline is used, the comparison with neutral would reduce noise associated specifically with the autobiographical recall required by the task (line 492).

Response #4:

We have followed the suggestion and further discussed the justification for using baseline to contrast emotional event to test predictions of PCT in the related section as follows:

Line 351: ". Finally, the contrast of interest was adjusted as emotion conditions were compared to either the neutral condition or the implicit baseline for BET and PCT respectively. This was in line with the account for the claim of PCT of that similar mechanisms support both neutral and emotional events. Contrasting emotional with neutral condition would then filter out any processes of interest including conceptualization, language, and executive attention serving to construct emotional events²."

Comment #5:

The discussion could explore in a little more detail the meaning and interpretation of the neural activity related to happiness (H3).

Response #5:

We have followed Reviewer #1's suggestion and further discussed the neural activity during the experience of happiness in the related section as follows:

Line 287: "However, for happiness, both events activated particularly the early visual cortices. This may be related to a vivid visual imagery of the happy events³. While overlapping activations for multiple events of a single emotion category are in line with BET predictions, the present topography is less so. Indeed, BET accounts associate the neural basis of happiness with the ACC² and a limited part of the present locations fall within the GAN or the meta-analytic map for positive affect⁴."

Reviewer #2

Comment #1:

It seems that within-emotion between-event conjunction/classification/RSA (H1-6) is done between functional runs, while between-emotion conjunction/classification/RSA (H7-H9) is done within functional runs. Is this correct? That makes a direct comparison between the two (H10) problematic, as higher similarity/ classification/ activation overlap in the present results could merely reflect the temporal distance between the events (see e.g., Mumford, J. A., Davis, T., & Poldrack, R. A. (2014). The impact of study design on pattern estimation for single-trial multivariate pattern analysis. *Neuroimage*, 103, 130-138). The solution to this problem might be simple: between-run analyses only. This might not leave enough events for SVM, but RSA is sufficient to test the hypotheses.

Response #1:

It seems there is a bit misunderstanding here. All the analyses were performed between functional runs not within runs. Between-emotion category specificity was tested at H1-2 as between-emotion category conjunction and classification analyses performed between functional runs. In H3-5 within-emotion category between-event conjunction, classification, and similarity (RSA) analyses performed between functional runs. Within-emotion category between-subjects similarity (RSA) analyses were performed between functional runs to test H6. To test between-emotion category consistency at H7-9, between-emotion category conjunction, correlation, and similarity (RSA) analyses across subjects were performed again between functional runs. However, following other suggestions from Reviewer #2 (see below) and Reviewer #3, we have dropped the H10 in the revised manuscript. Furthermore, we have added an additional table (line 147: Table 1: Specific hypotheses derived from emotion theories) in the revised manuscript to ease the understanding and to follow-up of hypothesis.

Comment #2:

The authors adjust the contrasts of interest based on the theories, which precludes direct comparisons of the theories. For within emotion category consistency, emotional memory retrieval is contrasted with neutral memory retrieval, properly control for non-emotional experiences (i.e., memory processing). Not much survives in the conjunction analyses and RSA values in the GAN are low, but significant. There could be a power issue, but the results do speak against stringent forms of BET. So far so good. But then for the across emotion category consistency emotional memory retrieval is contrasted with implicit baseline, which obviously reveals much higher correlations across emotions, because there is no control for the fact that participants are doing something cognitively demanding (so pattern similarity might be driven by this cognitive demand/ memory retrieval, nothing to do with emotions). It is not tested whether within-emotion category similarity explains any additional variance on top of the variance explained by between emotion-category. And this is necessary, because if the assumption is that the task is reliable, then a similar event should evoke similar BOLD patterns, at least as similar as the BOLD pattern related to a different event (would be best tested with Bayesian analyses). There is no theory that argues why between-emotions patterns should be higher than within-emotions between-events patterns, and it is unclear why the authors would have predicted this (H10).

Response #2:

We thank the Reviewer #2 for highlighting this issue. First, contrasting emotion conditions with neutral condition would lead us to lose conceptualization of PCT, which we discussed the justification further in the discussion as follows:
Line 351: “. Finally, the contrast of interest was adjusted as emotion conditions were compared to either the neutral condition or the implicit baseline for BET and PCT respectively. This was in line with the account for the claim of PCT of that similar mechanisms support both neutral and emotional events. Contrasting emotional with neutral condition would then filter out any

processes of interest including conceptualization, language, and executive attention serving to construct emotional events².” On the other hand, we agree with Reviewer #2 on direct comparison of the theories. So, we have changed our H5 from “The similarity between neural patterns across events within an emotion category is high/low (BET/PCT)” to “The similarity between neural patterns across events within an emotion category is higher than between emotion category (BET)” in the revised manuscript (line 124). To investigate this, we contrasted emotion condition with neutral condition and compared the within emotion correlations to between emotion correlations, testing whether within-emotion category similarity explains any additional variance beyond the variance explained by between emotion-category and enabling direct comparison. Finally, we agree with Reviewer #2 on H10 and now, we have left out the hypothesis 10 in the revised manuscript.
Comment #3: Relatedly, the absolute correlation values in RSA analyses shouldn’t be interpreted as ‘high’ or ‘low’ (even if point 1 & 2 are addressed). There are so many factors influencing these absolute values (e.g., filtering, power, specific task, specific contrast). fMRI is a noisy measure, and the design has very few events - in the end it is about whether there is any variance explained by within-emotion category similarity above and beyond between-emotion category. If so, that is evidence for (less stringent forms of) BET, if not, it supports PCT. But the bottom line is that the comparisons need to be done on the same contrast (either emo vs neutral, or emo vs baseline for both within and between emotion category similarity) and between functional runs only.
Response #3: We have changed the H5 as “The similarity between neural patterns across events within an emotion category is higher than between emotion category (BET)”. So, we tested whether the correlations within emotion category is significantly higher than between emotion categories, contrasting emotional conditions with neutral condition, which enables direct comparison. In addition, we have followed Reviewer #2’s suggestion and rephrased interpretation of RSA as ‘high’ or ‘low’ similarities instead of ‘high’ or ‘low’ correlations in the revised manuscript (line 216). Furthermore, we have dropped H10 in the revised manuscript.
Minor Comment #4: The autobiographical memory retrieval is used as an emotion induction. Although intensity does not significantly differ between emotional categories, variances didn’t seem equal (figure S1). Is there a reason why intensity is not included in the model, as regressor of no interest?
Response #4: We thank Reviewer #2 for pointing this out. To follow up on this, and we have re-analyzed the entire dataset, but now included with the intensity ratings as regressor weights in the GLM. This did not change the pattern of results.
Comment #5: If I understand it correctly, participants re-experienced their emotion for 61 seconds. It wasn’t quite clear to me from the description how this was modeled in the GLM. Is the entire minute modeled as one event?
Response #5: We clarified this, including the following section: Line 435: “The entire 61 seconds of emotion experience event was modelled as single event in the GLM.” in the revised manuscript.
Comment #6: Regarding the data availability: the authors state that the data cannot be archived publicly because of confidentiality. I assume this regards the autobiographical memory scripts, and this makes sense. However, there is no reason why the other data could not be made available (i.e., affect labels of the events / ratings scales / raw, defaced imaging data). It is the sharing of this

data that allows other researchers to reproduce results (data availability 'upon request' puts up unnecessary barriers to such attempts).

Response #6:

We totally agree with Reviewer #2 on reproducibility; however, we are not able to share the data since it is precluded by the local ethical committee. On the other hand, we have now plotted events and the subjects as individual datapoints in the intensity rating plot (Figure S1), shared more information on autobiographical events of the participants, and provided the source data underlying the graph as Supplementary Data 1 with the revised manuscript.

Reviewer #3

Major comments:

Comment #1:

The dataset used for the study is relatively small, $n=36$ (or 32 for some analyses) and only 2 instances of 4 emotions each. I wondered if this dataset is large enough (has enough power) to detect differences between emotion categories, similarities between events within a category, and whether there are enough datapoints for reliable SVM classification (which is based on a split dataset (1 event per category each) for training and testing). More importantly, I wondered if a possible lack of power perhaps biases the findings against BET and in favour of PCT, as BET predictions seem to be based on analyses of (pairwise) differences between categories or similarities between events within a category (small subsets of the data), whereas PCT predictions seem to be based on analyses across categories (all data). Could the authors provide a power calculation for the different analyses and explain whether this may be a problem or not (i.e., is there a possible lack of power and if so, could this differentially affect the support for the two theories (or why not)?

Response #1:

The median sample size for the most highly-cited fMRI studies published in high-impact journals is between 12-14⁵. The present sample size is thus more than twice this size. While this does not exclude a lack of power, it shows that the sample vastly exceeds the typical size of high-impact studies.

Comment #2:

The results state that intensity ratings did not differ between emotion categories, but there seems to be quite some variation within categories (figure S1), i.e., between subjects and events. I wondered to what extent this may have affected the conclusions of the study: If similar neural regions were activated, but with different intensity, how would that affect the different analyses (in particular conclusions with regards to (dis)similarity of neural patterns)? I think this is important as several analyses were based on within-category comparisons. Also, it would be helpful to see the variation between subjects and events plotted as individual datapoints in the figure (S1).

Response #2:

We thank Reviewer #3 for pointing this out. We have re-analyzed the entire dataset and included the intensity ratings as regressor weights in the GLM. This statistically accounts for the effect of intensity and the pattern of results did not change. Additionally, we have followed Reviewer #3's suggestion and plotted the subjects and events as individual datapoints in figure S1.

Minor comments

Comment #3:

It is not entirely clear to me, in the introduction, what the difference is between H3, H4, and H5. Don't they in fact test the same BET hypothesis (i.e., different events of the same category activate similar neural patterns), but perhaps only with different analysis methods?

Response #3: Indeed H3, H4, and H5 test relation between different events of the same emotion category. The specific difference between these are: H3 tests this at the level of the neural response amplitude, while H4 and H5 test this at the level of the neural response pattern (neural representation). H4 tests whether a classifier is able to classify emotion categories, if we train it using 1 of the 2 events per emotion category of every subject and test the classifier with the remaining events. We have changed H5 in the revised manuscript and it tests whether the neural pattern similarity within an emotion category is higher than between emotion category (line 124).
Comment #4: It is not clear to me why, according to PCT, the neural consistency is stronger across than within categories (H10, introduction line 136-137, and discussion line 330-331). Given theorized emotion-general elicitation mechanisms (across categories) and individual specificity (within categories), I can understand why PCT would perhaps posit similar consistency across as within categories. But why would the neural patterns be more similar for events that elicit different emotion experiences (consistency across categories) than for events with a similar emotion experience? Please explain.
Response #4: We thank the Reviewer #3 for pointing this out. We agree with Reviewer #3 and have dropped H10 in the revised manuscript.
Comment #5: Intensity ratings: It is not clear from the methods (line 384) and results (line 163) on which scale the ratings were performed, which makes it difficult to interpret these findings. Please add the scale range in the description.
Response #5: The visual analogous scale that is presented during the fMRI session was a slider which allows participants to rate their intensity of re-experience from 'very weak' to 'very intense'. Then the intensity ratings were calculated on a scale from 0 to 1 depending on the position of the slider as from 'very weak' to 'very intense'. To make it clearer, we have followed Reviewer #3's suggestion and added scale range in the description by rephrasing the following sections: Line 155: "The intensity ratings averaged over events of each emotion category ranged between .16 and 1 (on the scale of 0-1 as from 'very weak' to 'very intense')." Line 413: "First, the intensity ratings are calculated on a scale from 0 to 1 depending on the position of the slider (visual analogous scale) as from 'very weak' to 'very intense'. "Figure S1. Combined violin-box and whisker plots of the intensity ratings on a scale of 10, the subjects and events are plotted as individual datapoints for each emotion category."
Comment #6: The meaning of the results is often not clear (for a reader unfamiliar with the specific analyses). It would be helpful if the hypotheses were briefly repeated here (instead of only referring to the number, e.g., H1), and whether the result supported the hypothesis or not. E.g., line 177: "no significant results for any emotion category at voxel-wise whole brain level". Does this mean no similarities or no differences between categories? And is this in line with H1 or not? I find the following results similarly unclear (please explain what this means in reference to the respective hypothesis):  - Line 186: The trained classifier performed above chance, but only when taking the pattern in the GAN as a whole into account. - Line 197: No significant results for anger or sadness. - Line 197-198: significant clusters, but a limited part of these fell within the GAN. - Line 200: The classification did not perform above chance - Line 204 and 211: Significant yet low positive correlations - Examples where I did find the results clear are e.g. line 231: "analyses revealed extensive overlap across the entire brain", or line 234: "revealing consistent inter-individual activity associations"

between emotion categories over large portions of the brain". Please explain the other findings in a similar fashion.

Response #6:

We thank Reviewer #3 for pointing this out and have followed the Reviewer #3's suggestions, repeated the hypotheses and explained the interpretation of the results clearer, rephrasing following sentences:

Line 169: "In order to test H1 (each emotion category activates dedicated structures stronger than any other category (BET)), each emotion category was contrasted with the each of the other categories. There were no significant results for any emotion category at voxel-wise whole brain level, showing no emotion-category specific activation."

Line 179: "Both whole brain and ROI level results were in conflict with the prediction of BET relating to dedicated emotion category-specific neural circuits."

Line 180: "H2 (neural patterns have significant categorical diagnostic value (BET)) was tested by performing support vector machine (SVM) classification. The trained classifier performed above chance, but only when taking the pattern in the GAN as a whole into account (50.926% accuracy; $p < .01$) (Figure 2d), and not in any of the 36 local maxima, conflicting with the stringent variants of BET due to the lack of regional discriminatory information regarding the emotion category in the local maxima."

Line 193: "Within emotion category between event conjunction analyses were performed to test H3 (different events of a single emotion category activate similar structures (BET)). This revealed no significant results for anger or sadness, conflicting with the prediction of BET relating to consistency of different events of a single emotion category."

Line 196: "For happiness, significant clusters were located in bilateral occipital pole, cerebellar vermis, and mesencephalon (Figure 3), but a limited part of these clusters fell within the GAN. This result shows a neural consistency between different events of happiness as in line with BET, however the topography challenges BET since the consistent activations were mainly outside of the affect sensitive areas."

Line 200: "H4 (the neural pattern of one event is diagnostic for the neural pattern of another event of the same category (BET)) was tested by SVM classification analyses using training-independent dataset and the resulting classification did not perform significantly above chance in any of the 37 ROIs (GAN and its 36 spherical ROIs) (all p 's $> .01$). This suggests a lack of categorical discriminating consistency of activation patterns across events, compatible with PCT instead BET."

Line 204: "H5 (the similarity between neural patterns across events within an emotion category is higher than between emotion category (BET)) was tested by comparison of correlations across events within an emotion category and correlations across events between different emotion categories. One-tailed paired t-tests on the Fisher-transformed within category between event and between category between event Pearson correlation coefficients did not revealed any significant results in the GAN for any of the emotion categories (i.e., the between event within category correlations were not significantly greater than between event between category correlations) (all p 's $> .056$) (Figure 4), incompatible with BET."

Line 211: "H6 (the similarity between neural patterns across subjects is high/low (BET/PCT)) was tested by similarity analyses and the results revealed pairwise correlation coefficients ranging between $-.304$ and $.467$. This heterogeneous pattern was consistent in the GAN for each of the emotion categories (Figure 5a). One-tailed one-sample t-tests on the Fisher-transformed within category between subject Pearson correlation coefficients again showed significant, yet low similarity for each emotion category in the GAN (anger ($r = .033$; $t(629) = 7.222$, $p < .001$), sadness ($r = .018$; $t(629) = 4.277$, $p < .001$) and happiness ($r = .059$; $t(629) = 12.302$, $p < .001$)). The similarity analyses show some relevant consistency between events and across subjects within emotion category, mildly compatible with BET due to small effect sizes (all r 's $< .1$). The similarity analyses show some relevant consistency between events and across subjects within emotion category, mildly compatible with BET due to small effect sizes (all r 's $< .1$)."

Line 234: “Three pairwise between emotion conjunction analyses were performed to test H7 (the overlap in activation between emotion categories is high (PCT)). This revealed extensive overlap across the entire brain (Figure 6).”

Line 236: “H8 (there is a significant association between emotion categories across subjects (PCT)) was tested by between emotion category across subject correlation analyses and each of the 3 pairwise emotion category correlation analyses revealed widespread and extensive significant results, revealing consistent inter-individual activity associations between emotion categories over large portions of the brain (Figure 7).”

Line 240: “H9 (the pattern similarity across emotion categories is high/low (PCT/BET)) was tested using similarity analyses and the results revealed very low dissimilarities across all pairwise emotion category combinations (Figure 5b).”

Comment #7:

In the discussion, there is some ambiguity with regards to the extent to which some findings support BET or not, as there are more and less stringent formulations of BET. It seems that some findings do support some predictions of (less stringent versions of) BET:

- Line 276 (H2): “in line with predictions from less stringent variants of BET”.

- Line 288 (H3): “for happiness, both events activated the bilateral occipital pole, cerebellar vermis, and mesencephalon.”

- Line 300 (H4): “the patterns show some relevant consistency between events of a single category”.

- Line 303 (H6): “associations ranging between weakly negative and moderately positive”.

But the overall conclusions of the authors do not seem to really acknowledge that:

- Line 281: “Overall, these findings are incompatible with less stringent variants of BET”.

- Line 301: “the limited effect challenges essentialist BET predictions”.

- Line 306: “Findings are only mildly compatible with BET predictions”.

I wondered how predominant stringent vs. less stringent versions of BET are in the contemporary literature. Aren’t more modern versions less stringent? If that is so, I think that deserves acknowledgement in the discussion (e.g., results are not in line with original stringent BET, but do partially support (less stringent) modern versions).

Response #7:

Indeed, the less stringent versions of BET are the contemporary versions. We explained the essential differences between them in the introduction as follows:

Line 89: “BET predicts high specificity in neural architecture between distinct emotion categories⁶⁻⁸. Stringent and simplistic variants of BET propose a one-to-one mapping between emotion categories (e.g. anger) and activation in well-defined neural structures (e.g. orbitofrontal cortex, OFC). More contemporary BET variants acknowledge the modulation of distant regions by core structures as well as the importance of the neural pattern in addition to the intensity of the activation⁹. In this perspective, 12 out of 14 emotion categories including basic and non-basic emotions were distinguishable based on neural pattern, claiming that different emotions can be characterized by distinct neural signatures within a shared neural circuitry¹⁰.”

On the other hand, in this study, we focus on contrasting the stringiest BET with PCT.

Nevertheless, we have followed the Reviewer #3’s suggestion and discussed the results related to BET more nuanced as follows:

Line 274: “This is in line with previous findings⁹ and with predictions from less stringent variants of BET, which acknowledge that emotion categories are distinguishable based on distinct neural patterns within the shared common neural circuitry.”

Line 287: “However, for happiness, both events activated particularly the early visual cortices. This may be related to a vivid visual imagery of the happy events³. While overlapping activations for multiple events of a single emotion category are in line with BET predictions, the present topography is less so. Indeed, BET accounts associate the neural basis of happiness with the ACC²

and a limited part of the present locations fall within the GAN or the meta-analytic map for positive affect⁴.”

Line 300 was related to H5 and we have changed the hypothesis. The findings do not support any of the BET variants.

Line 281: “Overall, these findings are incompatible with stringent variants of BET as they are in conflict with the assumption of emotion category-specific dedicated neural circuits and locationism, and more compatible with less stringent variants of BET and PCT.”

We have removed line 301.

Line 305: “The findings are overall only mildly compatible with both variants of BET predictions, as these posit stronger across subject consistency, adhering to the assumption of a genetic basis and universality.”

Comment #8:

The selection of ROIs was based on a meta-analysis that included both studies that investigated emotion perception and studies investigating emotion experience (see line 435-436). However, according to the authors there may be important differences in emotional responses with these methods (see line 79-80). I wondered if/how the inclusion of perception studies in the ROI selection, while the focus of the current study was on experience, may have influenced the results.

Response #8:

According to our knowledge, there is no such a large meta-analysis (397 neuroimaging studies) considering solely emotion experience studies. Therefore, we considered the independent meta-analysis we used the most appropriate database.

Updated Figures

The following changes were made in the updated figures:

Figure 1: There was a problem with the overlay of the meta-analytic GAN on the inflated cortex in the initial submission. This is now fixed in the revised submission.

Figure 2-3, 5-7: We have re-analyzed the entire dataset and included the intensity ratings as regressor weights in the GLM based on suggestions of Reviewer #2 and Reviewer #3. Although the pattern of the results did not change, there were some minor changes in the extent and location of the significant clusters.

Figure 4: This figure corresponds to H5 that is tested in the study. We have changed our H5 from “The similarity between neural patterns across events within an emotion category is high/low (BET/PCT)” to “The similarity between neural patterns across events within an emotion category is higher than between emotion category (BET)” in the revised manuscript. So, the entire figure has changed.

Figure S1: The intensity ratings have plotted as individual datapoints for the subjects and events of each emotion in the revised manuscript based on the suggestion of Reviewer #3.

Figure 1. The meta-analytic general affect network (GAN), the affect-sensitive regions, and their overlap are shown on an inflated folded cortex.

Figure 2. Emotion category specific activations and classification performance. a-c: Combined violin-box and whisker plots of the beta values for anger-, sadness-, and happiness-specific activation in the OFC, ACC, and MPFC, respectively. $**V=489, p=.007$. d: Results of SVM classification of emotion categories for the GAN. Classification accuracy (red dot) superimposed on box and whisker plot following 200 permutation tests (50.926% accuracy; $p<.01$).

Figure 3. Statistical map representing result of conjunction analysis between happy events presented on coronal and sagittal slices ($p < .001$).

Figure 4. Combined violin-box and whisker plots of between event Pearson correlation coefficients (r) in the GAN.

Figure 5. Dissimilarity matrices showing the neural pattern dissimilarity using the distance metric (d), which is calculated by the equation ' $d=1-r$ ' where r is Pearson correlation coefficient, thus d values range from 0.0 (minimum distance) to 2.0 (maximum distance) with 1.0 (no correlation) in the middle. a: Dissimilarity matrices of the GAN, representing activation pattern dissimilarities within category between subjects for anger, sadness, and happiness. b: Activation-pattern dissimilarities across emotion categories within the GAN. $***t(35) > 21.884, p < .001$.

Figure 6. Probabilistic map of 3 pairwise between emotion category conjunction results represented on an inflated cortex.

Figure 7. Probabilistic map of 3 pairwise between emotion category correlation results represented on an inflated cortex.

Figure S1. Combined violin-box and whisker plots of the intensity ratings on a scale of 10, the subjects and events plotted as individual datapoints for each emotion.

References

- 1 Izard, C. E. Emotion theory and research: highlights, unanswered questions, and emerging issues. *Annu Rev Psychol* **60**, 1-25, doi:10.1146/annurev.psych.60.110707.163539 (2009).
- 2 Lindquist, K. A., Wager, T. D., Kober, H., Bliss-Moreau, E. & Barrett, L. F. The brain basis of emotion: a meta-analytic review. *Behav Brain Sci* **35**, 121-143, doi:10.1017/s0140525x11000446 (2012).
- 3 Dijkstra, N., Bosch, S. E. & van Gerven, M. A. Vividness of Visual Imagery Depends on the Neural Overlap with Perception in Visual Areas. *J Neurosci* **37**, 1367-1373, doi:10.1523/jneurosci.3022-16.2016 (2017).
- 4 Lindquist, K. A., Satpute, A. B., Wager, T. D., Weber, J. & Barrett, L. F. The Brain Basis of Positive and Negative Affect: Evidence from a Meta-Analysis of the Human Neuroimaging Literature. *Cereb Cortex* **26**, 1910-1922, doi:10.1093/cercor/bhv001 (2016).
- 5 Szucs, D. & Ioannidis, J. P. Sample size evolution in neuroimaging research: An evaluation of highly-cited studies (1990-2012) and of latest practices (2017-2018) in high-impact journals. *Neuroimage* **221**, 117164, doi:10.1016/j.neuroimage.2020.117164 (2020).
- 6 Izard, C. E. Forms and functions of emotions: Matters of emotion–cognition interactions. *Emotion review* **3**, 371-378 (2011).
- 7 Dalglish, T. & Power, M. *Handbook of cognition and emotion*. (John Wiley & Sons, 2000).
- 8 Panksepp, J. *Affective neuroscience: The foundations of human and animal emotions*. (Oxford university press, 2004).

- 9 Saarimäki, H. *et al.* Discrete Neural Signatures of Basic Emotions. *Cereb Cortex* **26**, 2563-2573, doi:10.1093/cercor/bhv086 (2016).
- 10 Saarimäki, H. *et al.* Distributed affective space represents multiple emotion categories across the human brain. *Soc Cogn Affect Neurosci* **13**, 471-482, doi:10.1093/scan/nsy018 (2018).

Reviewers' comments:

Reviewer #1 (Remarks to the Author):

This paper tests predictions generated by two emotion theories, Basic Emotions Theory (BET) and Psychological Construction Theory (PCT), and examines neurobiological mechanisms of emotion experience to determine which theory is supported. The study utilises an autobiographical memory recall task eliciting three different emotions (happy, sad, angry, and neutral). Results show neural activation patterns during emotional experiences are more consistent with predictions based on PCT than BET. Results indicate a common neural basis of affective experience, which is similar across the different emotion categories. The paper explores an interesting question and the approach is original. I think the changes made in response to the reviewer comments have clarified and strengthened this paper and I commend the authors on their work.

I have one small comment, I think that it would be useful to report whether the intensity of the emotion experiences differed between the two experiences of the same emotion. In other words, were both autobiographical recall experiences (e.g. of happy experiences) equally intense for participants?

Note, there is a typo on line 209 it should be reveal not revealed.

Reviewer #2 (Remarks to the Author):

The authors have addressed my first major concern (clarifying that classification was indeed only performed between-run) in both letter and manuscript. I also find the authors' conclusion that the data (specifically related to H1 & H2) speak against stringent forms of BET compelling. However, with regard to the other conclusions my second and third major concerns still stand.

The authors' explanation for using different baselines for testing different theories is unconvincing in the sense that it still makes a direct comparison of these theories problematic and that it undermines any claims based on H7-H9 as regarding the 'neurobiological basis of affect', unless PCT describes 'affect' as 'any cognitive activity'. But even if there is no way around using contrasts with implicit rather than the neutral event baseline (H7-H9), it is unclear why H9 is not tested by directly comparing within and between category similarity (as is done with regard to H5 - Figure 4 is excellent!). Again, as far as I can tell, the crucial question is whether within-emotion category similarity explains any additional variance on top of the variance explained by between emotion-category similarity, and this comparison can be made independent of the baseline that is used to extract the neural patterns of interest. A qualitative interpretation of similarity/ correlations as 'high' or 'low' (as the authors continue to do throughout) is problematic for reasons explained earlier and does not constitute a proper test of these theories.

Reviewer #3 (Remarks to the Author):

I would like to thank the authors for the effort of re-analyzing the data and adapting the figures to take into account the intensity ratings (major comment #2). This must have been a lot of work and I am happy to see that it did not substantially change the results and conclusions. Also the rewriting of the results section has made the findings much more clear. All in all, the authors have satisfactorily addressed most of my comments and I think the manuscript has improved.

Two of my comments were however not addressed to my satisfaction. As one of them (major comment #1) was my main concern with the manuscript, I would like to ask for a revision to address this concern. It would then also be nice to get a better answer to my minor comment #8, which was also not really addressed. Please find a short explanation of my remaining concerns below:

1. Major comment (original #1):

The authors do not respond to the content of the concern and basically dismiss it by stating that others have published papers with similar sample sizes. I find this very unsatisfactory, and I would like to see the concern itself addressed. Please note that my concern (lack of power, including possibility of biasing results towards one of the 2 theories) is not about the nr of participants or the fMRI method in general, but about the exceptionally low number of trials. I think this may be a limitation in general (as this could lead to less accurate effect estimates), but especially for the machine learning (SVM) method (which uses 1/2 of the data to train a model, and the other 1/2 to test it), which showed no significant discrimination (H2, p.8 line 180-185 and H4, p.10 line 200-204), i.e. did not perform above chance, which was taken as evidence against BET and consistent with PCT (see also discussion, p.16 line 295). My concern is that a) these results may be due to a lack of power (not enough data points for SVM method), and b) that moreover such a lack of power may bias the results against BET (vs PCT), because of how the different hypotheses were formulated (discrimination vs consistency) and tested (within vs. between category effects). In response, I would like the authors to either a) demonstrate that this is not a concern (i.e. provide support that the dataset is large enough for the SVM analyses (for both theories) or b) remove these analyses (if the data are insufficient for the method) or c) discuss this (lack of power and potential bias) as a limitation in the manuscript (if the data are borderline sufficient).

2. Minor comment (original #8):

The selection of ROI was based on a meta-analysis including both emotion perception and experience studies. I understand that this may have been the most appropriate database available, as experience studies are uncommon (as the authors explain in the introduction). However, as the authors also mention in the introduction that these two methods may lead to important differences in emotional responses (introduction p3 line 70-71), I would like them to explain the nature of these differences in more detail, especially with regards to the ROI. That is, to what extent are the different ROI based on/supported by perception vs experience research (esp. how much overlap or differences are there between the ROI when comparing these methods), and could it be that some of the ROI were not significant in the present study due to this difference in methods (i.e. ROI based mostly on perception studies and current study based on experience)? I would like to see the author's response to this question, and if this may have affected the results I would also like to see this included in the discussion of the paper.

Response to Reviewers

We thank the Reviewers for their constructive comments and their praise of the manuscript and the study as “The paper explores an interesting question and the approach is original. I think the changes made in response to the reviewer comments have clarified and strengthened this paper and I commend the authors on their work”, “I also find the authors’ conclusion that the data (specifically related to H1 & H2) speak against stringent forms of BET compelling”, “I would like to thank the authors for the effort of re-analyzing the data and adapting the figures to take into account the intensity ratings (major comment #2). This must have been a lot of work and I am happy to see that it did not substantially change the results and conclusions. Also the rewriting of the results section has made the findings much more clear. All in all, the authors have satisfactorily addressed most of my comments and I think the manuscript has improved”.

Reviewer #1

Comment #1:

I have one small comment, I think that it would be useful to report whether the intensity of the emotion experiences differed between the two experiences of the same emotion. In other words, were both autobiographical recall experiences (e.g. of happy experiences) equally intense for participants?

Response #1:

We thank Reviewer #1 for raising this issue and have followed up on the suggestion. We have reported this in the revised manuscript:

Line 166: “Furthermore, Shapiro-Wilk tests revealed that not all the intensity ratings across subjects within an event of an emotion were normally distributed (all p 's > .0005). Wilcoxon signed rank tests revealed that the intensity ratings of the emotion experience between events of the same emotion differed only for anger (anger: $p = .025$; sadness: $p = .368$; happiness: $p = .805$) (Figure 1).”

Line 498: “Furthermore, we tested whether the intensity ratings across subjects per event were normally distributed using Shapiro-Wilk tests. Then again depending on the normality, we tested whether the intensity ratings of two events of the same emotion category differed using parametric or non-parametric paired testing.”

We have accordingly updated Figure 1.

Reviewer #2

Comment #1:

The authors’ explanation for using different baselines for testing different theories is unconvincing in the sense that it still makes a direct comparison of these theories problematic and that it undermines any claims based on H7-H9 as regarding the ‘neurobiological basis of affect’, unless PCT describes ‘affect’ as ‘any cognitive activity’. But even if there is no way around using contrasts with implicit rather than the neutral event baseline (H7-H9), it is unclear why H9 is not tested by directly comparing within and between category similarity (as is done with regard to H5 - Figure 4 is excellent!). Again, as far as I can tell, the crucial question is whether within-emotion category similarity explains any additional variance on top of the variance explained by between emotion-category similarity, and this comparison can be made independent of the baseline that is used to extract the neural patterns of interest. A qualitative interpretation of similarity/ correlations as ‘high’ or ‘low’ (as the authors continue to do throughout) is problematic for reasons explained earlier and does not constitute a proper test of these theories.

Response #1:

We thank Reviewer #2 for the kind words about Figure 4. We have followed up on the comment and included the neutral condition as one of the conditions in addition to the emotional conditions for H7 and H8 (H5 and H6 in the revised manuscript), as PCT posits that similar mechanisms support both neutral and emotional events¹. We have updated the results and the figures (Figure 7,8 in the revised manuscript) and rephrased the following parts:

Line 264: "Six pairwise between emotion conjunction analyses were performed to test H5 (the overlap in activation between emotion categories is high (PCT)). This revealed extensive overlap across a large portion of the brain (Figure 7). H6 (there is a significant association between emotion categories across subjects (PCT)) was tested by performing between emotion category across subject correlation analyses for each of the 6 pairwise emotion category combinations. The resulting probability map revealed widespread significant results, revealing consistent inter-individual activity associations between emotion categories over large portions of the brain (Figure 8)."

Line 596: "We performed 6 voxel-wise whole-brain conjunction analyses between activation for each of the six emotion category pairs (anger-sad; anger-happy; anger-neutral; sad-happy; sad-neutral; happy-neutral). Each of the emotion conditions was compared to implicit baseline and not to the neutral condition, as PCT posits that similar mechanisms support both neutral and emotional events, in line with recent findings on emotion processing⁵⁴. We also investigated whether there were associations between emotion categories in the regional activation level across subjects by calculating voxel-wise whole-brain Pearson correlation coefficients between each of the six emotion category pairs (anger-sad; anger-happy; anger-neutral; sad-happy; sad-neutral; happy-neutral)."

Next, we have followed the suggestion of Reviewer #2, and tested H9 as H5 (H7 and H3, respectively in the revised manuscript). Furthermore, for both hypotheses we included analyses where we compared within vs between correlations combined over emotions in addition to emotion category specific comparisons. We have changed H9 from "The pattern similarity across emotion categories is high/low (PCT/BET)" to "H7: The similarity between neural patterns across events within an emotion category is not significantly different than between emotion categories (PCT)." and updated the figures (Figure 5, 9 in the revised manuscript), results, discussion, and the methods as follows:

Line 220: "H3 (the similarity between neural patterns across events within an emotion category is higher than between emotion category (BET)) was tested by comparing correlations across events within an emotion category with correlations across events between different emotion categories. Shapiro-Wilk tests revealed that all pairwise between event correlations were normally distributed (all p 's > .103). One-tailed (based on the BET prediction of higher correlations within emotion category) paired t-tests on the Fisher Z-transformed within category between event and between category between event Pearson correlation coefficients per emotion did not reveal any significant results in the GAN for any of the emotion categories (i.e., the between event within category correlations were not significantly stronger than the between event between category correlations for any of the emotion categories) (all $t(31)$'s < 1.635, all p 's > .056) (Figure 5a-c), incompatible with BET. Furthermore, Shapiro-Wilk tests revealed that both all combined within category correlations and all combined between category correlations were normally distributed (all p 's > .374). Following one-tailed two-sample t-test on all combined Fisher Z-transformed within category between event correlations and all combined between category between event Pearson correlations did not reveal any significant results in the GAN ($t(286) = 0.986$, $p = .162$) (Figure 5d)."

Line 271: "H7 (the similarity between neural patterns across events within an emotion category is not significantly different than between emotion categories (PCT)) was tested using similarity analyses. Shapiro-Wilk tests revealed that not all pairwise between emotion correlations were normally distributed (all p 's > .009), so non-parametric analyses were performed. Wilcoxon signed rank exact tests on the Fisher Z-transformed within category between event and between

category between event Pearson correlation coefficients showed significant results for anger-anger vs anger-sadness ($V=388$, $p=.019$) and anger-anger vs anger-happiness ($V=375$, $p=.038$) in the GAN (Figure 9a). However, the results did not reveal any other significant difference between within emotion combinations and between emotion combinations (all V 's < 304, p 's > .083) (Figure 9b,c). Furthermore, Shapiro-Wilk tests revealed that both all combined within category correlations ($p=.031$) and all combined between category correlations ($p<.001$) were not normally distributed. Wilcoxon signed rank test the all combined Fisher Z-transformed within emotion category between events and all combined between emotion category between events Pearson correlations did not revealed any significant result in the GAN ($V=264$, $p=.255$) (Figure 9d). The overall results are more compatible with PCT instead of BET."

Line 351: "Furthermore, predictions relating to within emotion category consistency were tested on the neural patterns. Between event similarity analyses (H3) revealed that pairwise correlations of different events within emotion category were significantly higher than the pairwise correlations of events of different emotion categories in the GAN, neither for any of the 3 emotion categories individually nor for all 3 emotion categories combined. This conflicts with one of the key predictions of BET on the consistency in activation patterns across events of a single emotion category while the findings can be explained by specificity and idiosyncrasy of each event in PCT."

Line 379: "Finally, across category consistency was tested at the neural representation level (H7). Pattern similarity analyses between events only revealed a significant difference in the pairwise correlation for anger-anger vs anger-sadness in the GAN but not for any other pairwise correlations per emotion or over all emotions combined. This result shows that the similarity between neural patterns across events within an emotion category is not significantly different than between emotion category. The strong correlations between all pairwise emotion category combinations (all r 's > .6) thus indicates consistent trans-categorical activation patterns, consistent with PCT predictions, based on the hypothesis of psychological primitives shared in all emotion categories supported by large-scale networks. Furthermore, this result puts the within category between event association in perspective, as it indicates that the significant between event within category association is not limited to within category conditions, but extends across categories."

Line 565: "Furthermore, to investigate the similarity of the neural patterns within emotion category between events, we performed similarity analyses between 1 of the 2 events of each emotion category and the remaining events of each emotion category (e.g. similarity analyses between anger vs neutral in the first run and anger vs neutral in the second run, between anger vs neutral in the first run and sadness vs neutral in the second run). In order to investigate whether the pattern similarity across events within an emotion category is higher than between emotion category, for each emotion category, pairwise Fisher Z-transformed Pearson correlation coefficients between events within emotion category were compared to pairwise Fisher Z-transformed Pearson correlation coefficient between events of that emotion category and events of other emotion categories (e.g. whether the correlations between anger vs neutral in the first run and anger vs neutral in the second run is significantly higher than correlations between anger vs neutral in the first run and sadness vs neutral in the second run). First, the normality of the distribution of pairwise Fisher Z-transformed Pearson correlation coefficients between events was tested for each variable individually, pooled within emotion correlations, and pooled between emotion correlations by means of Shapiro-Wilk tests. Depending on the results, the assessment of the significance was performed by means of 2 parametric or non-parametric paired testing on sets of 32 within subject between event Fisher Z-transformed correlations for each emotion category separately and by parametric or non-parametric testing on pooled within emotion correlations on the one hand and pooled between emotion correlations on the other hand."

Line 603: "In addition, similarity analyses between events were performed exactly in the same approach as the similarity analyses between events to test within emotion category consistency but emotion conditions were compared to implicit baseline instead of to the neutral condition. In order to investigate whether within emotion between event correlations are different than the

between emotion category between event correlations, the assessment of the significance was performed again as explained above. The similarity analysis was performed in the GAN only, in line with PCT hypothesis”

We have followed the suggestion of Reviewer #2 and rephrased the sentences referring to the interpretation of the similarity/correlations as follows:

We have rephrased H6 (H4 in the revised manuscript): “The similarity between neural patterns across subjects within an emotion category is significant (BET)”

Line 240: “One-tailed one-sample t-tests on the Fisher-transformed within category between subject Pearson correlation coefficients showed significant correlations, yet with small effect sizes for each emotion category in the GAN (anger ($r=.033$; $t(629)=7.222$, $p<.001$), sadness ($r=.018$; $t(629)=4.277$, $p<.001$) and happiness ($r=.059$; $t(629)=12.302$, $p<.001$)).”

Reviewer #3

Comment #1:

The authors do not respond to the content of the concern and basically dismiss it by stating that others have published papers with similar sample sizes. I find this very unsatisfactory, and I would like to see the concern itself addressed. Please note that my concern (lack of power, including possibility of biasing results towards one of the 2 theories) is not about the nr of participants or the fMRI method in general, but about the exceptionally low number of trials. I think this may be a limitation in general (as this could lead to less accurate effect estimates), but especially for the machine learning (SVM) method (which uses 1/2 of the data to train a model, and the other 1/2 to test it), which showed no significant discrimination (H2, p.8 line 180-185 and H4, p.10 line 200-204), i.e. did not perform above chance, which was taken as evidence against BET and consistent with PCT (see also discussion, p.16 line 295). My concern is that a) these results may be due to a lack of power (not enough data points for SVM method), and b) that moreover such a lack of power may bias the results against BET (vs PCT), because of how the different hypotheses were formulated (discrimination vs consistency) and tested (within vs. between category effects). In response, I would like the authors to either a) demonstrate that this is not a concern (i.e. provide support that the dataset is large enough for the SVM analyses (for both theories) or b) remove these analyses (if the data are insufficient for the method) or c) discuss this (lack of power and potential bias) as a limitation in the manuscript (if the data are borderline sufficient).

Response #1:

We have followed the suggestion of Reviewer #3 (option b) and the editorial office (disqualifying option c) and removed the SVM analyses (H2 and H4) in the revised manuscript.

Comment #2:

The selection of ROI was based on a meta-analysis including both emotion perception and experience studies. I understand that this may have been the most appropriate database available, as experience studies are uncommon (as the authors explain in the introduction). However, as the authors also mention in the introduction that these two methods may lead to important differences in emotional responses (introduction p3 line 70-71), I would like them to explain the nature of these differences in more detail, especially with regards to the ROI. That is, to what extent are the different ROI based on/supported by perception vs experience research (esp. how much overlap or differences are there between the ROI when comparing these methods), and could it be that some of the ROI were not significant in the present study due to this difference in methods (i.e. ROI based mostly on perception studies and current study based on experience)? I would like the author's response to this question, and if this may have affected the results I would also like to see this included in the discussion of the paper.

Response #2:

We have followed the suggestion of Reviewer #3 and included following part in the limitation section of the discussion of the revised manuscript:

Line 413: “Next, the independent meta-analysis we used in order to define the GAN includes emotion experience studies as well as emotion perception studies. Yet there are significant qualitative differences between the perception and experience of emotions. Emotion perception essentially reflects sensory (typically visual) processing of the external environment, which is often objectively standardized across participants as they are all shown the same stimuli. Emotion experience is a typically subjective process that has long been associated with processing of bodily sensations². The basis of the GAN, including both perception and experience studies, does thus not constitute the ideal one for the present purposes as it may over- and underemphasize perceptual and experiential regions respectively. Indeed, comparing the affect-sensitive regions defined in our dataset with the GAN, reveals that some regions, e.g. the somatosensory cortices, show emotional modulation in the present dataset, but not in the meta-analytic map. Remarkably, we also observed emotional modulation in early visual regions outside the meta-analytic map. Of note, the participants had their eyes closed during the emotion experience event. We presume this may be explained by visual imagery effects³. The comparison between our affect-sensitive map and the meta-analytic map reveals that there are relevant affect-sensitive regions outside the GAN, despite the liberal threshold we applied to the GAN. However, as we opted for an independently defined GAN, we considered this one to be the most appropriate database currently available, as experience studies are uncommon. Furthermore, the GAN covers the key results of seminal emotion experience studies⁴.”

Updated Figures

The following changes were made in the updated figures:

Figure 1: We have now plotted functional runs separately for each emotion category and reported whether the intensity ratings differed between the two events of the same emotion as suggested by Reviewer #1.

Figure 3: The SVM analyses and its plot has removed (H2 in the previous version of the manuscript). The plots correspond to H1 were converted into raincloud plots.

Figure 5: We have added panel d which represents the comparison between all combined between event within emotion correlations and all combined between event between emotion category correlations, relating to H3. In addition, all the plots were converted into raincloud plots.

Figure 6: Panel b has removed, which was relating to H7 (H9 in the previous version of the manuscript). It is presented in Figure 9.

Figure 7-8: We have included the neutral condition as one of the conditions in addition to the emotional conditions for H5 and H6 and created the probability maps again.

Figure 9: We have changed H9 from “The pattern similarity across emotion categories is high/low (PCT/BET)” to “H7: The similarity between neural patterns across events within an emotion category is not significantly different than between emotion categories (PCT). Correspondingly, an entire new figure has created by means of raincloud plots.

Figure 1. Combined raincloud-box and whisker plots of the intensity ratings on a scale of 10, the subjects plotted as individual datapoints for each emotional event. * $V=321.5$, $p=.025$.

Figure 3. Emotion category specific activations. a-c: Combined raincloud-box and whisker plots of the beta values for anger-, sadness-, and happiness-specific activation in the OFC, MPFC, and ACC, respectively. * $V=489$, $p=.007$.

Figure 5. Combined raincloud-box and whisker plots of between event Pearson correlation coefficients (r) in the GAN. a-d: Anger, sadness, happiness, and all emotions combined, respectively.

Figure 6. Dissimilarity matrices of the GAN, representing the neural pattern dissimilarities across subjects within category for anger, sadness, and happiness. Dissimilarity matrices were generated using the distance metric (d) which is calculated by the equation ' $d=1-r$ ' where r is Pearson correlation coefficient, thus d values range from 0.0 (minimum distance) to 2.0 (maximum distance) with 1.0 (no correlation) in the middle.

Figure 7. Probabilistic map of 6 pairwise between emotion category conjunction results represented on an inflated cortex.

Figure 8. Probabilistic map of 6 pairwise between emotion category correlation results represented on an inflated cortex.

Figure 9. Combined raincloud-box and whisker plots of between event Pearson correlation coefficients (r) in the GAN. a-d: Anger, sadness, happiness, and all emotions combined, respectively. * $V < 389$, $p < .05$.

References

- 1 Sokolov, A. A. *et al.* Brain circuits signaling the absence of emotion in body language. *Proc Natl Acad Sci U S A* **117**, 20868-20873, doi:10.1073/pnas.2007141117 (2020).
- 2 James, W. & Lange, C. G. *The Emotions; Volume I.* (Creative Media Partners, LLC, 2018).
- 3 Le Bihan, D. *et al.* Activation of human primary visual cortex during visual recall: a magnetic resonance imaging study. *Proc Natl Acad Sci U S A* **90**, 11802-11805, doi:10.1073/pnas.90.24.11802 (1993).
- 4 Damasio, A. R. *et al.* Subcortical and cortical brain activity during the feeling of self-generated emotions. *Nat Neurosci* **3**, 1049-1056, doi:10.1038/79871 (2000).

Reviewers' comments:

Reviewer #1 (Remarks to the Author):

This paper examines commonalities and specificities in neural activity during emotional experience to evaluate whether Basic Emotions Theory (BET) or Psychological Construction Theory (PCT) is better supported by the data. Using an autobiographical memory recall task participants experience three different emotions (happy, sad and angry, and a neutral experience) while fMRI is recorded. Results generally indicate that neural activation aligns with predictions derived from PCT compared to BET. I commend the authors on the work and modifications they have undertaken in response to reviewer comments, and am happy that they have satisfactorily addressed my comments.

Reviewer #2 (Remarks to the Author):

The manuscript has improved by removal of the SVM analyses and inclusion of the analyses depicted in Figure 9 (very clear!), comparing within category to between category similarity.

The first part of the paper is strong (H1-3), but I regret to say I'm still not convinced by the second part. Most importantly, the authors continue to interpret the strength of neural pattern correlations (rather than differences in similarity between relevant conditions, i.e., a subtractive methodology), e.g., "The strong correlations .. ($r > .6$) ... indicates consistent trans-categorical activation patterns, consistent with PCT predictions." (line 385). The idea that absolute representational similarity values can be used this way is already evident from the way some of the hypotheses are rephrased:

"H4: The similarity between neural patterns across subjects within an emotion category is significant (BET)" and "H6: There is a significant association between emotion categories across subjects (PCT)"

With RSA (as employed since Haxby et al., 2001), it is not relevant whether correlations are "high" or "low" or whether they are significant or not (similarity could be driven by all kinds of factors, especially without a meaningful baseline, i.e., H6): what matters is whether within category correlations are significantly higher than between category correlations. If not, this is evidence against BET, and there would be no need to interpret data as "mildly compatible with BET" (line 245). If there is, this suggests some uniqueness in representations and therefore support for BET. This is a straightforward test, i.e., Figure 6 would present one big RDM with pairwise correlations within and between category events, with the question being whether the within category correlations on average exceed the between category correlations. Alternatively, these analyses could be removed altogether.

I suppose I'm also still not following the rationale for analyses with an implicit baseline - testing whether 'task engagement' (rather than (emotional) event retrieval specifically) results in overlapping activity across events/ people (would be shocking if it wouldn't) - so I don't really see the point of H5-H7. But perhaps other emotion researchers do, and as long as within-category is contrasted with between-category similarity (as in tests of H3 and H7), the analyses itself are valid.

Some minor comments:

- It would be helpful to clarify the difference between Figure 9 (with implicit baseline) and Figure 5 (same analysis but with neutral baseline) in the figure legends. Both figures are excellent though, very clear way to present the data.
- The way H7 is phrased "H7: The similarity between neural patterns across events within an emotion category is not significantly different than between emotion categories" seems like this is the scenario in which H3 is rejected. It can be useful to test for evidence of 'no difference', but that really requires Bayesian testing. If the crucial difference with H3 is the use of different baselines (neutral vs implicit), make that clear in the hypothesis itself, and either phrase it from the BET perspective (expecting a higher similarity for within category than between category

similarity), or use Bayesian stats to get an idea of how strong the evidence is for the null hypothesis.

- Figure legends of Figure 7 and 8: What values are presented here? Probability of what?

Reviewer #3 (Remarks to the Author):

The authors have satisfactorily addressed my 2 remaining comments.

Only with the track changes it wasn't clear to me which Figures were added to or removed from the manuscript. If Fig 3 on page 11 and Fig 5 on page 15 are the old ones (replaced by the figures on page 10 and 14 above, respectively) then I think it's ok. If not, there appears to be an inconsistency between the notes and the figures regarding panel d in both cases.

Response to Reviewers

We thank the Reviewers for their constructive comments and their praise of the manuscript and the study as “I commend the authors on the work and modifications they have undertaken in response to reviewer comments, and am happy that they have satisfactorily addressed my comments”, “The manuscript has improved by removal of the SVM analyses and inclusion of the analyses depicted in Figure 9 (very clear!), comparing within category to between category similarity”, and “The authors have satisfactorily addressed my 2 remaining comments”.

Reviewer #2

Comment #1:

The first part of the paper is strong (H1-3), but I regret to say I’m still not convinced by the second part. Most importantly, the authors continue to interpret the strength of neural pattern correlations (rather than differences in similarity between relevant conditions, i.e., a subtractive methodology), e.g., “The strong correlations .. ($r > .6$) ... indicates consistent trans-categorical activation patterns, consistent with PCT predictions.”(line 385). The idea that absolute representational similarity values can be used this way is already evident from the way some of the hypotheses are rephrased:

“H4: The similarity between neural patterns across subjects within an emotion category is significant (BET)” and “H6: There is a significant association between emotion categories across subjects (PCT)”

With RSA (as employed since Haxby et al., 2001), it is not relevant whether correlations are “high” or “low” or whether they are significant or not (similarity could be driven by all kinds of factors, especially without a meaningful baseline, i.e., H6): what matters is whether within category correlations are significantly higher than between category correlations. If not, this is evidence against BET, and there would be no need to interpret data as “mildly compatible with BET” (line 245). If there is, this suggests some uniqueness in representations and therefore support for BET. This is a straightforward test, i.e., Figure 6 would present one big RDM with pairwise correlations within and between category events, with the question being whether the within category correlations on average exceed the between category correlations. Alternatively, these analyses could be removed altogether.

I suppose I’m also still not following the rationale for analyses with an implicit baseline - testing whether ‘task engagement’ (rather than (emotional) event retrieval specifically) results in overlapping activity across events/ people (would be shocking if it wouldn’t) – so I don’t really see the point of H5-H7. But perhaps other emotion researchers do, and as long as within-category is contrasted with between-category similarity (as in tests of H3 and H7), the analyses itself are valid.

Response #1:

We thank reviewer #2 for the kind words about H1-3. We have removed all the parts that interpret RSA results as strength of neural pattern correlations and used a comparative phrasing between frameworks. Regarding H4, we have followed Reviewer#2’s suggestion and presented one big RDM with pairwise correlations within and between category events. We rephrased the hypothesis as “H5: The similarity between neural patterns across subjects within an emotion category is significantly higher than between emotion category (BET)” and calculated whether the within category correlations on average exceed the between category correlations.

In addition, we have added another hypothesis that relates to consistency over subjects in regional activation, instead of over patterns. This new hypothesis precedes the one described above: “H4: There is a significant overlap in regional activation across subjects within an emotion category (BET)”. For this purpose, we first identified for each subject the regions that were active during each emotion category vs neutral at a liberal threshold ($p < .05$, uncorrected) and subsequently created maps displaying percentage overlap across subjects for each emotion category. Furthermore, we calculated the maximal percentage of subjects to obtain a minimal overlap. We have updated the results, discussion, and methods as follows:

“H4 (there is a significant overlap in regional activation across subjects within an emotion category (BET)) was tested by calculating the maximal percentage of subjects to obtain a minimal overlap in emotion category specific activation. First, we defined emotion-specific activation at a liberal threshold ($p < .05$, uncorrected) at subject-level. The resulting statistical map was binarized and probability maps across subjects were then computed. These revealed that each of 7 subjects (19%) showed activation in a cluster located in the culmen of the left cerebellum during anger experience (Figure S1), 6 subjects (16%) each activated two clusters located in right thalamus and the declive of right cerebellum during experience of sadness (Figure S2), and each of 7 subjects (19%) activated a cluster located in left primary visual cortex during the experience of happiness (Figure S3). The limited overlap ($< 20\%$ for each emotion category) and the topography challenge the universal characteristics of basic emotions and the predefined brain-emotion associations by BET. Furthermore, H5 (the similarity between neural patterns across subjects within an emotion category is significantly higher than between emotion category (BET)) was tested by comparing correlations across subjects within an emotion category with correlations across subjects between emotion category. Similarity analyses revealed a heterogeneous pattern (Figure 6a). Interestingly, the dissimilarity matrix clearly shows decreased dissimilarity within subject between emotion categories, compared to both between subject within emotion category and between subject between emotion category. The former reflects idiosyncratic across category neural patterns, in line with PCT. Shapiro-Wilk tests revealed that all pairwise between subject correlations were normally distributed (all p 's $> .154$), so parametric testing performed. F tests revealed there was a difference in variance between sadness-sadness vs sadness-happiness ($p = .012$). One-tailed Welch two sample t-tests revealed significant results for anger-anger vs anger-sadness ($t(1196.4) = 2.138$, $p = .016$), happiness-happiness vs anger-happiness ($t(1240.9) = 5.664$, $p < .001$), and happiness-happiness vs sadness-happiness ($t(1213.4) = 6.16$, $p < .001$) (Figure 6b). However, the results did not reveal any significant result for anger-anger vs anger-happiness and for sadness-sadness vs any other combination (all t 's < 1.241 , all p 's $> .107$). In addition, Shapiro-Wilk tests revealed that both pooled within category across subject correlations and pooled between category across subject correlations were not normally distributed (all p 's $< .01$). One-tailed Wilcoxon signed rank test showed significant results for pooled within category across subject correlations vs pooled between category across subject correlations ($W = 3781486$, $p < .001$) (Figure 6c). These results show consistency between neural patterns across subjects for happiness and pooled across within emotion category correlations, in line with BET, however not for the categories anger and sadness.”

“Furthermore, within emotion category between subjects conjunction analyses (H4) revealed very limited spatial overlap between subjects ($< 20\%$ of all subjects) within each emotion category. The findings are only mildly compatible with both variants of BET predictions, as these posit strong across subject consistency, adhering to the assumption of a genetic basis and universality.”

“Next, within emotion consistency was tested across subjects in regional activation and neural patterns. Probability maps of subject overlap in binarized regional within emotion category activation maps (H4) revealed very limited spatial overlap between subjects ($< 20\%$ of all subjects) within each emotion category. Furthermore, predictions relating to within emotion neural pattern consistency across subjects (H5) were tested using similarity analyses. The results revealed that for 1 of the 3 categories (i.e. happiness) and for all categories combined, pairwise correlations across subjects within emotion were significantly higher than for any between emotion category

combination. The finding that 2 out of 3 categories did not show this effect, conflicts with BET, positing strong across subject consistency, adhering to the assumption of a genetic basis and universality.”

“Next, to investigate spatial overlap between subjects within emotion categories, we first identified for each subject the regions that were active during each emotion category vs neutral at a liberal threshold ($p < .05$, uncorrected), biasing the methods in favour of BET, and subsequently created binary maps, of which we calculated the percentage overlap across subjects for each emotion category. We then calculated the maximal percentage of subjects to obtain a minimal overlap. Furthermore, to investigate the similarity of neural patterns within emotion categories across subjects, we performed similarity analyses across subjects for within each emotion category and between each emotion category pair, revealing all 630 (for each within emotion category) and 1260 (for each between emotion) pairwise between subject Pearson correlations of the neural patterns in the GAN³⁷. In order to investigate whether the pattern similarity across subjects within an emotion category is higher than between emotion category, we compared pairwise Fisher Z-transformed Pearson correlation coefficients between subjects within emotion category to pairwise Fisher Z-transformed Pearson correlation coefficients between subjects between emotion category for each emotion category separately and all emotion categories combined (pooled within emotion categories vs pooled between emotion categories). First, the normality of the distribution of all pairwise Fisher Z-transformed Pearson correlation coefficients was tested using Shapiro-Wilk tests for each variable individually, pooled within emotion correlations, and pooled between emotion correlations. Depending on the normality, the assessment of pattern similarity significance was performed by means of 2 parametric or non-parametric paired testing on all pairwise Fisher Z-transformed Pearson correlation coefficients for every emotion category separately and by parametric or non-parametric testing on pooled within emotion correlations and pooled between emotion categories.”

Regarding H6, the hypothesis has been tested by calculating voxel-wise whole-brain Pearson correlation coefficients in regional activation not by RSA, as mentioned in the manuscript:

“We also investigated whether there were associations between emotion categories in the regional activation level across subjects by calculating voxel-wise whole-brain Pearson correlation coefficients between each of the six emotion category pairs (anger-sad; anger-happy; anger-neutral; sad-happy; sad-neutral; happy-neutral).”

Minor

Comment #2:

It would be helpful to clarify the difference between Figure 9 (with implicit baseline) and Figure 5 (same analysis but with neutral baseline) in the figure legends. Both figures are excellent though, very clear way to present the data.

Response #2:

We thank Reviewer #2 for the kind words and pointing this uncertainty about Figure 5 and 9 and. We have rephrased the legends of Figure 5 and 9:

“Figure 5. Combined raincloud-box and whisker plots of between event Pearson correlation coefficients (r) in the GAN. a-d: Anger vs neutral, sadness vs neutral, happiness vs neutral, and all emotions combined, respectively.”

“Figure 9. Combined raincloud-box and whisker plots of between event Pearson correlation coefficients (r) in the GAN. a-d: Anger vs baseline, sadness vs baseline, happiness vs baseline, and all emotions combined, respectively. * $V=375$, $p=.019$; ** $V=388$, $p=.010$.”

Comment #3:

The way H7 is phrased “H7: The similarity between neural patterns across events within an emotion category is not significantly different than between emotion categories” seems like this is the scenario in which H3 is rejected. It can be useful to test for evidence of ‘no difference’, but that really requires Bayesian testing. If the crucial difference with H3 is the use of different baselines (neutral vs implicit), make that clear in the hypothesis itself, and either phrase it from

the BET perspective (expecting a higher similarity for within category than between category similarity), or use Bayesian stats to get an idea of how strong the evidence is for the null hypothesis.

Response #3:

We have followed suggestion of Reviewer #2 and rephrased H3 and H7 (H8 in the revised manuscript) as follows:

“H3: The similarity between neural patterns across events within an emotion category (vs neutral) is significantly higher than between emotion category (vs neutral) (BET)”

“H8: The similarity between neural patterns across events within an emotion category (vs baseline) is significantly higher than between emotion categories (vs baseline) (BET)”

Comment #4:

Figure legends of Figure 7 and 8: What values are presented here? Probability of what?

Response #4:

We thank Reviewer #2 for pointing this uncertainty out. Probabilistic functional maps represent spatial consistency by showing the significant task activity of the relative number of subjects or conditions at each spatial location. We have made this clear in the figure legends of Figure 7 and 8 by rephrasing them:

“Figure 7. Probabilistic map of spatial overlap of 6 pairwise between emotion category conjunction results represented on an inflated cortex.”

“Figure 8. Probabilistic map of spatial overlap of 6 pairwise between emotion category correlation results represented on an inflated cortex.”

Reviewer #3

Comment #1:

Only with the track changes it wasn't clear to me which Figures were added to or removed from the manuscript. If Fig 3 on page 11 and Fig 5 on page 15 are the old ones (replaced by the figures on page 10 and 14 above, respectively) then I think it's ok. If not, there appears to be an inconsistency between the notes and the figures regarding panel d in both cases.

Response #1:

Yes, indeed Fig 3 on page 11 and Fig 5 on page 15 have been replaced by the figures on page 10 and on page 14, respectively.

Updated Figures

The following changes were made in the updated figures:

Figure 6: We have presented one big RDM with pairwise correlations within and between category events as suggested by Reviewer #2. We also added new raincloud plots relating to “H5: The similarity between neural patterns across subjects within an emotion category is significantly higher than between emotion category (BET)”.

Figure 9: By changing “H7: The similarity between neural patterns across events within an emotion category is not significantly different than between emotion categories (PCT)” to “H8: The similarity between neural patterns across events within an emotion category (vs baseline) is significantly higher than between emotion categories (vs baseline) (BET)”, there has been a change in the significance on the plot.

Figure 6. Within emotion category consistency across subjects. a: The dissimilarity matrix of the GAN, representing the neural pattern dissimilarities across subjects within and between emotion category. The dissimilarity matrix was generated using the distance metric (d) which is calculated by the equation ' $d=1-r$ ' where r is Pearson correlation coefficient, thus d values range from 0.0 (minimum distance) to 2.0 (maximum distance) with 1.0 (no correlation) in the middle. 's' represents the subject numbers. b-c: Combined raincloud-box and whisker plots of between subject Pearson correlation coefficients (r) in the GAN, per emotion and all emotions combined, respectively. $*t(1196.4)=2.1384, p=.016$; $***p<.001$.

Figure 9. Combined raincloud-box and whisker plots of between event Pearson correlation coefficients (r) in the GAN. a-d: Anger vs baseline, sadness vs baseline, happiness vs baseline, and all emotions combined, respectively. $*V=375, p=.019$; $**V=388, p=.010$.